# The burden of malaria-attributable maternal anaemia and the impact of preventive treatment across sub-Saharan Africa

**A list of authors and their affiliations appears at the end of the paper**

Malaria in pregnancy is a major but poorly quantified contributor to maternal anaemia in sub-Saharan Africa. We combined individual-level data on haemoglobin (Hb), gravidity, gestational age and PCR-confirmed *Plasmodium falciparum* infection from 12,608 pregnancies in 7 African countries with a gravidity-specific model of malaria exposure and immunity linked to contemporary maps of transmission and fertility. For 2023, we estimate that 13.1 million pregnancies in malaria-endemic African regions were exposed to *P. falciparum*. In the absence of preventive measures, this exposure would have resulted in 2.41 million (95% credible interval 1.98–3.04 million) cases of moderate or severe anaemia (Hb < 9 g dl$^{-1}$), including 600,000 (408,000–906,000) severe cases (Hb < 7 g dl$^{-1}$). A counterfactual scenario using 2,000 transmission levels suggests that a 32% reduction in exposure during pregnancy translated into only a 22% decline in intrinsic anaemia burden, reflecting a shift from a concentration of risk in primigravidae to a more even distribution across gravidities as multigravid women acquire less pregnancy-specific immunity. Calibrating our model to randomized trials, we estimate that under current coverage, intermittent preventive treatment of malaria in pregnancy using sulfadoxine-pyrimethamine averted around 1.10 million (0.72–1.61 million) cases of moderate or severe anaemia and 330,000 (225,000–523,000) severe cases in 2023. These findings show that although burden has declined substantially, malaria remains a major driver of maternal anaemia risk. Meanwhile, lower immunity across multigravidae means any interruption to intermittent preventive treatment of malaria in pregnancy using sulfadoxine-pyrimethamine, or other population-based malaria control efforts, risks rapid resurgence of severe maternal anaemia, with substantial consequences for maternal and neonatal health.

After major declines in the early twenty-first century, progress in malaria control has stalled amid chronic underinvestment[1]. The World Health Organization (WHO) estimates that there were 263 million malaria cases globally in 2023, 22 million more than in 2013, causing 597,000 deaths[2]. These gains are at risk amid major reductions in global health funding[3]. Historically, disruptions to malaria control programmes have led to rapid and severe rebounds in transmission[4], driven by the parasite's short generation time and mosquito vectors' 1–2 week lifespan[5].

Global estimates of the burden of malaria often exclude the consequences of malaria upon maternal health. In 2023, an estimated 12.4 million pregnancies were exposed to *Plasmodium falciparum* in 34 moderate-to-high-transmission countries within the WHO African Region, increasing the risk of adverse pregnancy outcomes[2,6]. During pregnancy, the parasite expresses a variant surface antigen that enables binding to chondroitin sulfate A (CSA) receptors abundant in the placental intervillous space[7], facilitating sequestration of infected

✉e-mail: patrick.walker@imperial.ac.uk

**Fig. 1 | The effect of malaria infection on Hb dynamics during pregnancy.**
**a**, Posterior estimates (yellow lines) of the reduction in Hb resulting from malaria infection among women experiencing their first malaria-exposed pregnancy, including primigravidae and multigravidae without prior exposure. The dots and the error bars show the observed differences in Hb between infected and uninfected primigravidae (the only group in these data for whom absence of prior malaria exposure in pregnancy can be unambiguously established) grouped according to study-specific quintiles of gestational age at enrolment and plotted at the mean of each quintile. **b**, Estimated average reduction in Hb (g dl⁻¹) at 140 days' gestation as a function of the number of previous malaria-exposed pregnancies. **c**, Comparison of model-based estimates (lines) and observed differences (points) in Hb at 140 days' gestation as a function of primigravid PCR prevalence of *P. falciparum*, stratified according to gravidity. The model estimates shown are for the fertility pattern used to represent each trial site with the median total fertility rate (TFR) (Southern Malawi from the 2017 Malaria Indicator Survey (MIS)); the dots and error bars represent trial-specific means and 95% CIs for the difference in Hb between infected and uninfected women enrolled between 120 and 160 days' gestation. In all plots, the thin lines show 1,000 draws from the joint posterior distribution of the fitted model; the thick lines indicate the posterior medians and 95% Wald CIs computed using a Welch-type standard error (allowing for different sample sizes and variances in the infected and uninfected groups). In **a** and **c**, data from studies excluding women with severe anaemia at enrolment are coloured blue, those including all women regardless of haemoglobin status are coloured orange.

erythrocytes, often at very high densities[8]. This placental sequestration increases the risk of low birthweight (LBW), preterm birth and maternal anaemia[9]. To mitigate these risks, WHO recommends insecticide-treated nets (ITNs), intermittent preventive treatment of malaria in pregnancy using sulfadoxine-pyrimethamine (IPTp-SP) beginning in the second trimester, and effective case management[10]. However, coverage remains suboptimal and heterogeneous: in 2023, in the WHO African Region, only 59% of pregnant women reported ITN use and 44% received three or more IPTp-SP doses. Consequently, although IPTp-SP is estimated to have prevented 551,000 LBW cases, an additional 351,000 cases remained unprevented[2,6].

Malaria infection during pregnancy contributes to maternal anaemia, through placental sequestration of parasites, haemolysis, splenic clearance of infected erythrocytes and suppression of erythropoiesis[11–14]. Anaemia is associated with an increased risk of LBW, preterm birth and perinatal mortality, with risk escalating as haemoglobin (Hb) decreases[15]. Maternal anaemia is a major contributor to postpartum haemorrhage and maternal death: women with an Hb < 9 g dl⁻¹ have a threefold increase in the odds of postpartum haemorrhage[15]. In 2015, approximately 40% of maternal deaths in sub-Saharan Africa were due to haemorrhage; severe anaemia (Hb < 7 g dl⁻¹) is associated with a two-fold to threefold increase in the odds of maternal mortality compared to non-severe anaemia[15–17].

Although the link between malaria and maternal anaemia is well established, quantitative estimates of the malaria-attributable burden remain scarce and outdated. A 2007 literature review estimated that 26% of severe maternal anaemia cases (Hb < 7 g dl⁻¹ or Hb < 8 g dl⁻¹) were attributable to malaria[18]. However, this estimate was based on data collected before large-scale declines in malaria prevalence resulting from expanded treatment and vector control. Furthermore, substantial heterogeneity in malaria transmission across Africa[19] means that the true population-attributable fraction of anaemia due to malaria will vary widely. The relationship between malaria endemicity and anaemia burden is further complicated by the unique immunology of malaria in pregnancy. Women acquire an adaptive, antibody-mediated immune response that reduces placental sequestration of parasites in successive pregnancies[13,20]. This leads to a plateauing of malaria-attributable LBW risk at high-transmission levels, with the greatest burden concentrated among young, first-time mothers and their infants[21,22]. A similar pattern is likely for malaria-related anaemia but remains underquantified.

In this study, we aimed to estimate the burden of maternal anaemia across Africa attributable to malaria, while accounting for gravidity-specific immunity. We modelled the effect of malaria on Hb throughout gestation using individual-level data from clinical trials conducted across seven countries. This framework was linked to an existing mathematical model of malaria exposure during pregnancy as a function of population-level endemicity and fertility[23], accounting for the acquisition of immunity over successive infected pregnancies. Using these estimates alongside fine-resolution spatial data on malaria prevalence, population density and fertility patterns, we extrapolated the risk of anaemia in pregnancy across Africa according to different grades of

severity, based on Hb concentration thresholds. We first estimated the 'intrinsic risk', defined as the burden of malaria-attributable anaemia that would occur in the absence of pregnancy-specific interventions such as IPTp-SP. We also examined how intrinsic risk has changed over time in response to declining population-level transmission following the scale-up of malaria control since the early 2000s. We also examined how intrinsic risk has changed over time in response to declining population-level transmission after the scale-up of malaria control since the early 2000s. We then assessed how IPTp is currently mitigating this risk, providing updated impact estimates amid uncertain future global support for malaria control and maternal health.

## Results

### The effect of malaria on Hb dynamics during pregnancy

We analysed individual-level data on Hb concentration, gestational age and gravidity at first antenatal care (ANC1), from pregnant women enrolled in large-scale malaria intervention trials[24–27] conducted across a range of endemicity settings (10–58% *P. falciparum* polymerase chain reaction (PCR) prevalence among primigravid women at enrolment; details in Supplementary Table 1). Gestational age spanned 50–238 days (interquartile range of 126–165).

Several patterns were evident in the raw data (Fig. 1 and Extended Data Fig. 1). Among primigravid women, Hb was consistently lower in those infected with *P. falciparum* and showed a clear decline with advancing gestational age even in the absence of infection. The median Hb level in uninfected third-trimester primigravidae across studies (excluding those with anaemia-specific exclusion criteria) was 10.7 g dl[−1], meaning that many women already fell below the WHO-defined less than 10 g dl[−1] threshold for moderate anaemia[28]. Therefore, we report 'moderate-to-severe anaemia' as less than 9 g dl[−1], a threshold more strongly associated with postpartum haemorrhage risk[15,29], which better captures the incremental effect of malaria among women within a higher quartile of risk.

Both maternal age and gravidity were associated with progressively smaller malaria-attributable reductions in Hb in univariate models (Extended Data Fig. 2a,b), although these covariates were themselves highly correlated (Extended Data Fig. 2c). To disentangle their contributions, we fitted multivariable mixed-effects models, including both covariates, adjusting for gestational age with a natural cubic spline and a random intercept for study site. Adding maternal age improved model fit, which is consistent with higher baseline Hb at older ages; however, the interaction between malaria and age (controlling for gravidity) was minimal (Extended Data Fig. 2d). By contrast, the gravidity effect remained strong across age quintiles (Extended Data Fig. 2e), while gravidity-specific estimates changed minimally when age was added to the model (Extended Data Fig. 2f).

Despite the clear association with gravidity, the extent to which malaria was associated with Hb reduction varied across trial sites: in low-transmission settings, infected multigravidae still showed appreciable declines, whereas in high-transmission settings the effect was no longer detectable by the third pregnancy (Extended Data Fig. 1). By contrast, in primigravid women the reduction in Hb associated with infection was far more consistent across trials and showed no clear relationship with transmission intensity (Fig. 1c). These patterns are consistent with prior evidence from placental histopathology and immuno-epidemiological studies indicating that pregnancy-specific immunity—which protects against placental sequestration of parasites—is acquired progressively over successive infected pregnancies[11,30], with placental malaria-associated antibody acquisition showing an increased gradient according to gravidity with increasing transmission intensity[31]. However, as in most settings, prior malaria exposure during pregnancy was not directly observable.

To quantify this protective effect, we developed an inferential framework that estimated prior exposure using a previously developed model linked to G1 malaria prevalence (as measured using PCR) as a proxy for local transmission intensity[6,23]. This allowed us to estimate the extent of acquired immunity as a function of inferred exposure history, rather than relying on gravidity alone. We compared our exposure-based model to simpler alternatives that assumed Hb dynamics varied only according to gravidity or that infection effects were constant. The exposure-informed model provided a better fit to the observed data (Supplementary Table 2), supporting the use of transmission-calibrated exposure histories to capture immunity-dependent effects on Hb. This model indicated that in uninfected women, early pregnancy Hb levels are higher in multigravidae than in primigravidae (Supplementary Table 3) and decline steadily across all gravidity groups during the second trimester (Extended Data Fig. 1). In primigravid women, *P. falciparum* malaria infection was associated with a progressive reduction in Hb across gestation, from a decline of 1.31 g dl[−1] (95% credible interval (CrI) 0.75–1.86) at 80 days (late first trimester) to 1.82 g dl[−1] (95% CrI 1.41–2.22) by 200 days (early third trimester) (Fig. 1a). We estimated that by 140 days' gestation (mid-second trimester) infected multigravid women who had experienced a single prior infected pregnancy exhibited a much smaller reduction in Hb than primigravidae and multigravidae without prior exposure: 0.35 g dl[−1] (95% CrI 0.18–0.51) compared to a 1.38 g dl[−1] (95% CrI 1.25–1.51). This corresponds to a 74.8% (95% CrI 66.2–85.4%) mitigation of malaria-related Hb reduction because of immunity acquired from a first infected pregnancy. By the third infected pregnancy, we estimated that the negative effect of infection was nearly completely mitigated, with only a 0.01 g dl[−1] reduction (95% CrI 0.00–0.14) in Hb (Fig. 1b).

### The effect of malaria on infected pregnant women and population-level risk of anaemia

Figure 2a–d shows the risk per 1,000 infected pregnancies (top row, ordered according to gravidity category: G1, G2, G3+, all women) and risk per 1,000 total pregnancies (Fig. 2e–h, bottom row, in the same order). In women experiencing malaria infection during pregnancy for the first time, we estimated that the risk of moderate-to-severe anaemia rises sharply over the course of gestation (Fig. 2a), with a proportionally steeper rise in the risk of severe anaemia (Hb < 7 g dl[−1]). The modelled increase among women experiencing malaria infection during pregnancy for the first time spans from 189 (95% CrI 145–237) cases of moderate-to-severe anaemia and 17.5 (95% CrI 11.6–25.3) cases of severe anaemia per 1,000 infected pregnancies at 100 days' gestation (early second trimester), to 442 (95% CrI 355–530) and 124 (95% CrI 85.5–180) per 1,000, respectively, by 200 days (early third trimester). The population-level risk of anaemia among primigravidae increased monotonically with malaria transmission intensity (Fig. 2e). However, the average anaemia risk across all pregnancies is modulated by parity patterns and fertility rates. In settings with higher transmission intensity, the per-pregnancy anaemia risk among infected women is lower, driven by acquisition of immunity in multigravidae exposed to malaria in prior pregnancies (Fig. 2b,c). Thus, we observed a non-linear relationship between malaria exposure and anaemia risk, with per-pregnancy risk plateauing at high-transmission levels. This saturation effect was more pronounced in higher-fertility settings, where the average gravidity of pregnant women was greater than in lower-fertility regions (Figs. 2d,h and 3).

### Intrinsic malaria-attributable burden and progress since 2000

By integrating our Hb-based risk model with a spatially explicit transmission model, we estimated the continent-wide burden of malaria-attributable maternal anaemia across malaria-endemic regions of Africa, accounting for local variation in transmission intensity and fertility patterns (Fig. 4). We estimate that among the 41.8 million women who became pregnant in malaria-endemic regions of Africa in 2023, 13.1 million (95% CrI 12.4–14.0) were exposed to *P. falciparum* during pregnancy. Based on our model, this exposure caused an average

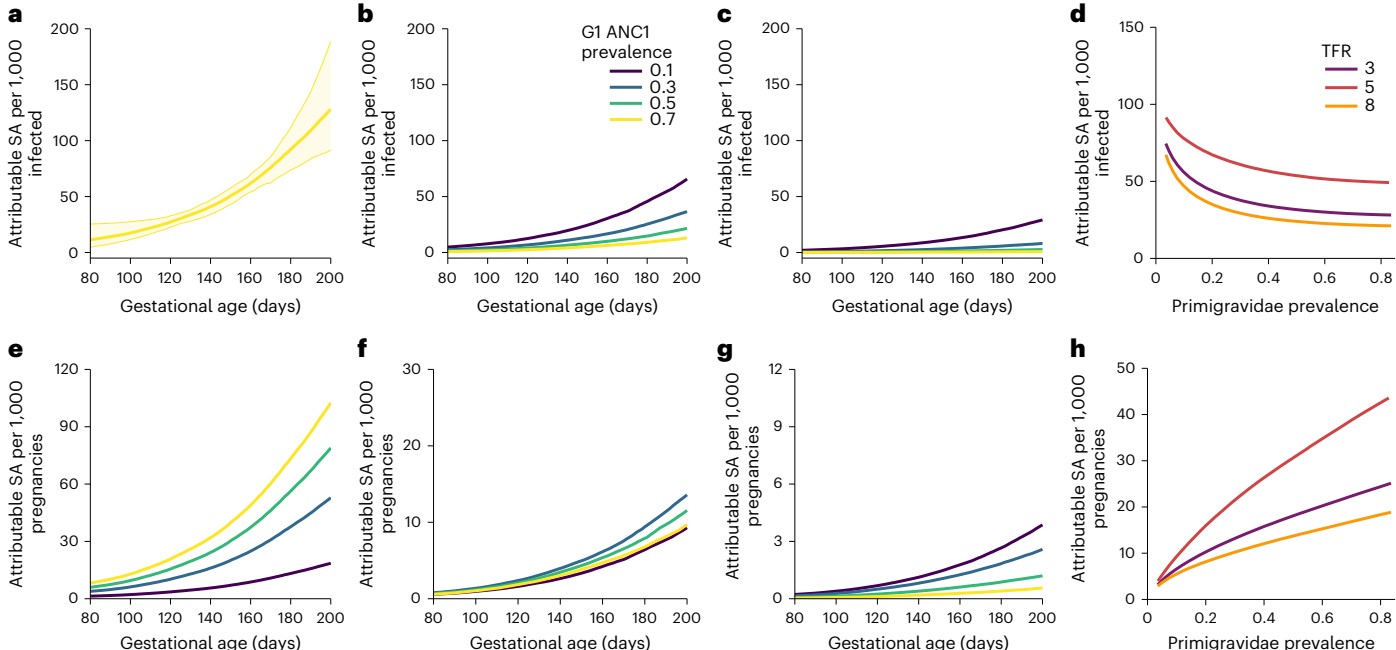

**Fig. 2 | The risk of severe anaemia attributable to malaria according to gestational age, gravidity and transmission intensity. a–c**, Estimated individual-level risk of severe anaemia (SA) attributable to malaria infection, defined as additional cases per 1,000 infected pregnancies, shown according to gestational age for primigravidae (G1; this does not vary according to transmission intensity) (**a**), secundigravidae (G2) (**b**) and gravida 3+ (G3+) (**c**) under varying levels of PCR prevalence among primigravidae at first ANC visit (ANC1). **d**, Individual-level attributable risk according to malaria PCR prevalence in primigravidae averaged across all gravidities using gravidity distributions from three example fertility settings: TFR 3, urban Kenya (2020 Kenya MIS[52]); TFR 5, southern Malawi (2017 Malawi MIS[53]); and TFR 8, rural Niger (2021 Niger MIS[54]). **e–h**, Corresponding population-level risk of severe anaemia attributable to malaria infection among 1,000 pregnancies for the same subgroups and settings as in **a** (**e**), **b** (**f**), **c** (**g**) and **d** (**h**), weighted according to the proportion of women infected at each gestational age and gravidity. **a**, The thick line shows the posterior median and the ribbons show the 95% CrI. **b–h**, Only posterior medians are shown for visual clarity.

reduction of 0.14 g dl⁻¹ (95% CrI 0.12–0.17) in Hb concentration by the beginning of the second trimester (100 days' gestation), that is, before women becoming eligible for intermittent preventive treatment in pregnancy (IPTp). This reduction in Hb translated to an estimated 766,000 (95% CrI 608,000–918,000) cases of moderate-to-severe anaemia affecting 1.83% (95% CrI 1.45–2.19%) of all pregnancies and 65,400 (95% CrI 43,700–86,400) cases of severe anaemia, affecting 0.16% (95% CrI 0.10–0.21%) of all pregnancies.

In the absence of effective preventive measures in the second trimester, we estimate that the average reduction in Hb concentration by the beginning of the third trimester (200 days' gestation) would have reached 0.25 g dl⁻¹ (95% CrI 0.21–0.31). The number of cases increases further when accounting for physiological Hb declines that also occur during gestation among uninfected women. By the beginning of the third trimester, we estimate 2.41 million (95% CrI 1.98–3.04) cases of moderate-to-severe anaemia, including 600,000 (95% CrI 408,000–906,000) severe cases; relative to the burden present at the beginning of the second trimester, this represents a 3.22-fold (95% CrI 2.65–4.23) and a 9.49-fold (95% CrI 6.74–14.8) increase, respectively.

Two countries, Nigeria and the Democratic Republic of the Congo (DRC), accounted together for 38.0% (95% CrI 36.0–39.6%) of the total potential burden of malaria-attributable maternal anaemia in 2023. The average per-pregnancy risk of anaemia within a country is largely shaped by a combination of transmission intensity and fertility rates, which are often negatively correlated. Together, these influence both the level of exposure and the degree of acquired immunity among pregnant women. For example, the DRC has the highest estimated per-pregnancy intrinsic risk of severe anaemia in primigravidae, at 72.1 per 1,000 pregnancies (95% CrI 43.2–115). However, the high fertility rate in this country results in a larger share of pregnancies occurring among multigravid women with substantial acquired immunity. Thus,

the overall per-pregnancy risk of severe anaemia in the DRC is estimated at 16.9 per 1,000 pregnancies (95% CrI 11.6–26.3), which is lower than in several countries with lower fertility rates (15 countries have point estimates for the average per-pregnancy risk of severe anaemia between 19 and 21 per 1,000 pregnancies).

To quantify the extent to which reductions in malaria transmission in the twenty-first century have mitigated the risk of maternal anaemia through decreased exposure during pregnancy, we estimated the number of malaria-attributable anaemia cases that would have occurred in a counterfactual scenario in which pregnancies in 2023 experienced transmission intensities observed in 2000, before the scale-up of large-scale interventions. We used *P. falciparum* prevalence in 2000 estimated by the Malaria Atlas Project (MAP). Under this counterfactual (Fig. 4), we estimate that 19.2 million (95% CrI 18.8–19.9 million) pregnant women would have been exposed to malaria; relative to the observed exposures in 2023, this reflects a 32.0% (95% CrI 26.8–36.4%) increase (Table 1).

However, the corresponding effect on the overall burden of anaemia has been more modest. We estimate that, under 2000 transmission conditions, 3.04 million (95% CrI 1.47–3.85 million) women would have experienced moderate-to-severe anaemia, of which 766,000 (95% CrI 536,000–1.15 million) would have been severe anaemia (Table 1 and Supplementary Table 4). Therefore, by comparison with our 2023 estimates without effective preventive measures, this suggests that the scale-up of population-level interventions averted 656,000 (95% CrI 442,000–930,000) moderate-to-severe anaemia cases (including 170,000 severe cases; 95% CrI 105,000–290,000), corresponding to relative reductions of 21.3% (95% CrI 16.9–24.9%) and 22.2% (95% CrI 17.7–26.0%), respectively.

Moreover, we estimate that there have been substantial shifts in the underlying epidemiological landscape, with an increasing majority of malaria in pregnancy occurring in mesoendemic or hypoendemic

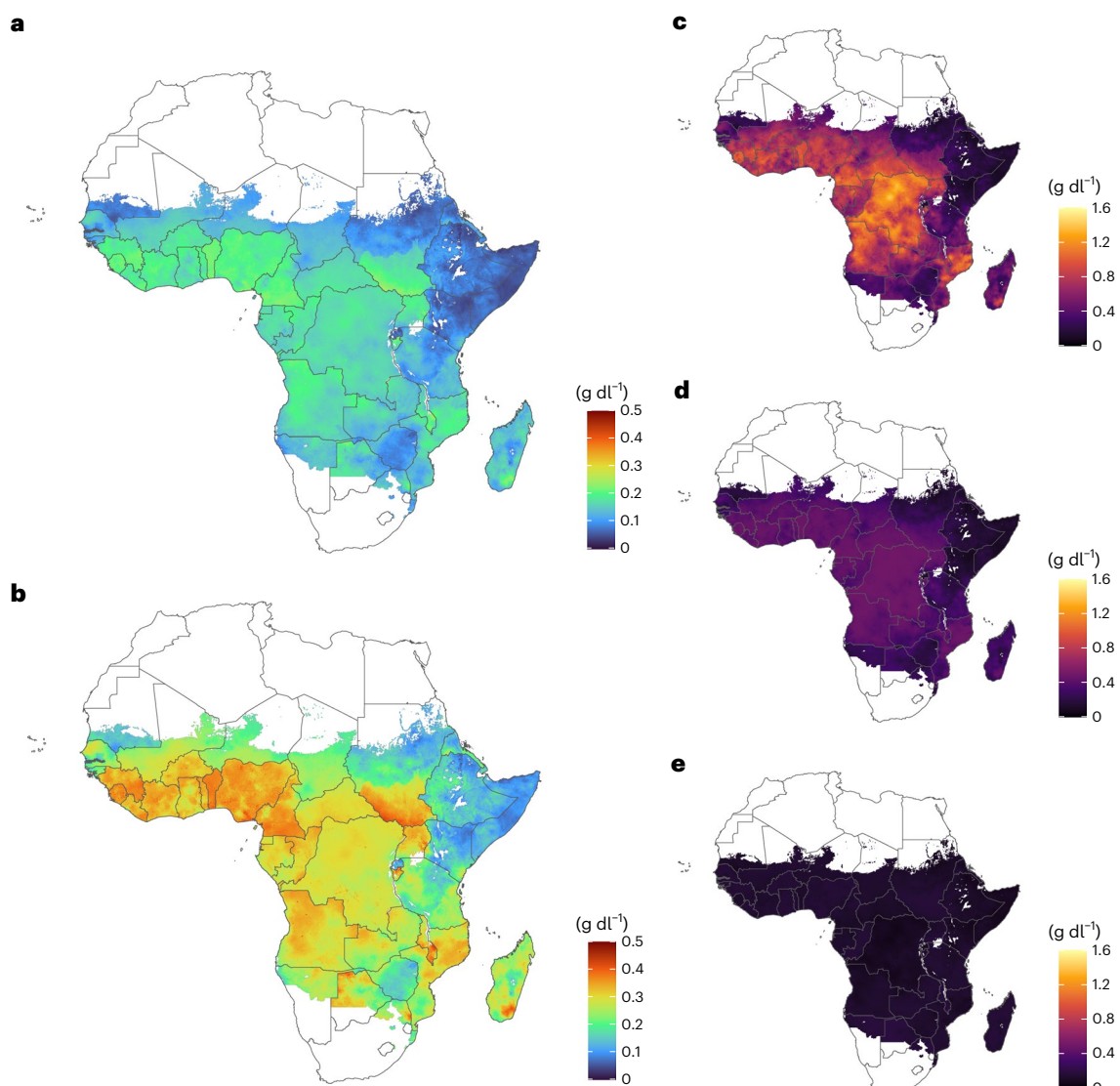

**Fig. 3 | Model-estimated malaria-attributable reduction in Hb concentration throughout gestation, stratified according to gravidity, in malaria-endemic countries in sub-Saharan Africa. a–e**, Estimated malaria-attributable reduction in Hb for all gravidities at the beginning of the second trimester (100 days' gestation) (**a**), all gravidities at the beginning of the third trimester (200 days' gestation) (**b**), primigravidae at the beginning of the third trimester (**c**), secundigravidae at the beginning of the third trimester (**d**) and women in their third pregnancy and beyond at the beginning of the third trimester (**e**). Administrative boundaries were obtained from GADM v.3.6.

settings (that is, areas with *P. falciparum* prevalence in 2–10-year-olds below 50%). Under the 2000 transmission levels, most of the malaria-attributable anaemia burden would have occurred among primigravidae: 67.2% (95% CrI 65.4–69.6%) of severe cases, equivalent to 521,000 (95% CrI 353,000–755,000) pregnant women and 58.8% (95% CrI 55.8–61.6%) of moderate-to-severe cases, totalling 1.79 million (95% CrI 1.47–3.85 million) women. By contrast, lower malaria exposure among primigravidae in recent years has led to reduced immunity in multigravid women and consequently a higher risk of anaemia per exposure. Thus, the overall reduction in severe malaria-attributable anaemia, 170,000 cases (95% CrI 105,000–290,000), is effectively entirely driven by reductions among primigravidae, whose burden fell by an estimated 179,000 cases (95% CrI 108,000–297,000), whereas the burden among multigravidae remained approximately stable, with a point estimate of a small increase of 8,100 cases (95% CrI −3,000 to 25,100) (Fig. 4b).

### Estimating the effect of IPTp on anaemia

Our finding that malaria has limited effect on Hb levels from the third infected pregnancy onwards is supported by randomized,

placebo-controlled trials of IPTp that have primarily been conducted in highly endemic settings[28,32–36]. These trials have consistently shown substantial improvements in Hb levels among paucigravidae (primigravidae and secundigravidae) who received IPTp, but no detectable effect in higher gravidities (Fig. 5 and Supplementary Table 4). To quantify IPTp-SP efficacy using these data, we calibrated our model to the baseline infection prevalence of each trial and estimated the malaria-attributable Hb reduction in each gravidity group. We then assumed that IPTp-SP mitigates this reduction by a constant proportion across settings, such that the absolute benefit is greatest in primigravidae and declines with increasing gravidity as the malaria-attributable effect itself becomes smaller. This model fitted the observed trial results well and yielded a maximum likelihood estimate of IPTp-SP efficacy: a 53.7% reduction (95% confidence interval (CI) 29.9–95.8%) in Hb loss that would otherwise occur because of malaria.

Extrapolating these effects to our estimates of Hb reductions and the resulting anaemia risk in the absence of IPTp, we estimate that universal IPTp-SP coverage across malaria-endemic regions of Africa could have averted approximately 1.45 million (95% CrI 0.93–2.27 million)

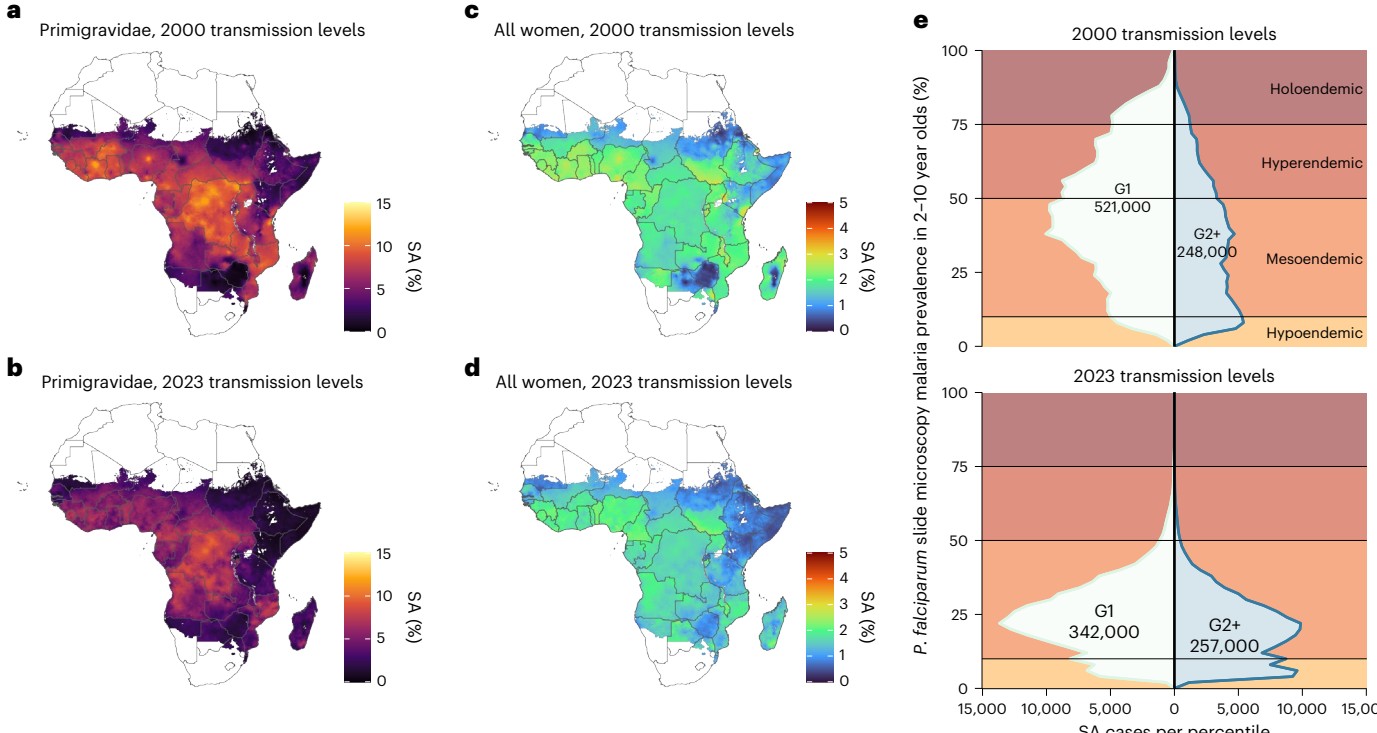

**Fig. 4 | Shifts in the landscape of malaria-attributable anaemia after changes in transmission since 2000. a–d**, Modelled maps of the intrinsic risk of malaria-attributable severe anaemia (Hb < 7 g dl⁻¹) across Africa, shown for 2023 under two scenarios: observed transmission levels based on *P. falciparum* prevalence in children aged 2–10 years (PfPr2–10, **b**,**d**) and a counterfactual scenario in which prevalence remained at the 2000 levels (**a**,**c**). Left, maps show the risk among primigravidae. Right: risk averaged across all gravidities. **e**, Distributions of total intrinsic burden stratified according to malaria transmission intensity (*y* axis)

and gravidity based on 2000 (top) and 2023 (bottom) transmission levels. In both plots, the region to the left of the origin represents the estimated frequency of intrinsic burden in primigravidae. The region to the right of the origin denotes the equivalent in multigravidae. Classical endemicity categories are shaded in red and orange for reference: hypoendemic (<10% PfPr2–10); mesoendemic (10–50%); hyperendemic (50–75%) and holoendemic (>75%). Administrative boundaries were obtained from GADM v.3.6.

cases of moderate-to-severe anaemia in 2023, including 458,000 (95% CrI 299,000–742,000) severe cases (Fig. 5). However, based on the 2023 IPTp-SP coverage estimates across 34 implementing countries, with 55% of pregnancies receiving at least two doses and 67% receiving at least one[2] (Methods), we estimate that the intervention averted 1.10 million (95% CrI 0.72–1.61 million) cases of moderate-to-severe anaemia (including 330,000 (95% CrI 225,000–523,000) severe cases) in practice (Table 1).

Overall, we estimate that the combination of IPTp-SP uptake and declines in community-level transmission since 2000 prevented 1.73 million (95% CrI 1.25–2.43 million) cases of moderate-to-severe maternal anaemia in 2023, of which 491,000 (324,000–793,000) were severe (Table 1 and Extended Data Table 1). Proportional reductions were broadly consistent across WHO African subregions, with point estimates ranging from 56.1% to 60.7% for moderate-to-severe anaemia and 67.0% to 71.9% for severe cases alone. However, these averages obscure important regional variation in the underlying drivers of impact (Fig. 5d,e). In East Africa, for example, reductions in community-level transmission mean that the intrinsic burden has declined more substantially than in other regions, but the proportional impact of IPTp-SP has been lower, driven in part by the fact that four countries (Ethiopia, Eritrea, Djibouti and Rwanda) do not currently implement IPTp-SP. We also stratified our analysis according to whether countries were designated by WHO as high burden to high impact (HBHI) focus countries, a classification used to prioritize malaria control efforts in the ten African countries with the highest absolute malaria burden[37]. We estimate that a disproportionately high number of anaemia cases were averted in these countries, driven primarily by higher-than-average IPTp-SP uptake. By contrast, declines in intrinsic burden have been greater

in non-HBHI countries, which is consistent with broader declines in malaria transmission across these lower-burden settings.

## Discussion

Despite substantial reductions in malaria transmission over the twenty-first century, malaria in pregnancy remains a leading contributor to a woman's risk of maternal anaemia and its consequences in much of sub-Saharan Africa. We estimate that in 2023, 13.1 million pregnancies were exposed to malaria[2,6]; in the absence of pregnancy-specific preventive measures, this would have caused over 2.4 million cases of moderate-to-severe maternal anaemia, including 600,000 severe cases.

Overall, we estimate that two key drivers, declining community-level malaria transmission and the continued scale-up of IPTp-SP, together prevented 1.73 million cases (95% CrI 1.25–2.43 million) of moderate-to-severe maternal anaemia in 2023, including 491,000 (324,000–793,000) severe cases. IPTp-SP remains the most widely recommended strategy to reduce malaria burden[10,32,38,39] and has been estimated as the most cost-effective of all major malaria interventions to prevent malaria burden generally[40]. Our estimates suggest that IPTp-SP mitigates malaria-attributable Hb loss by 53.7% (95% CI 29.9–95.8%). Across subregions, proportional reductions in malaria-attributable anaemia were consistent—around 60% for moderate-to-severe cases and 70% for severe cases alone. However, the relative contributions of transmission decline and IPTp-SP varied. In East Africa, reductions were primarily driven by lower malaria exposure since 2000, while in the ten HBHI countries, where intrinsic burden remains high, the greater-than-average reduction in malaria-attributable anaemia burden reflects higher IPTp-SP uptake. These findings highlight the importance of sustaining IPTp-SP

**Table 1 | Changes in intrinsic burden of severe maternal anaemia due to reductions in malaria transmission since 2000 and the effect of IPTp-SP**

| Area | Pregnancies at risk | | Percentage pregnancies exposed | | Intrinsic burden (×10³) | | Intrinsic burden (per 1,000 pregnancies) | | Percentage burden in primigravidae | | Incremental IPTp impact (×10³) | | Burden averted | |
|---|---|---|---|---|---|---|---|---|---|---|---|---|---|---|
| | Total (×10⁶) | G1% | 2000 | 2023 | 2000 | 2023 | Overall | G1 | 2000 | 2023 | Averted | Remaining | Total | % |
| All countries | 41.8 | 22.6 | 50.3 (49.1–52.0) | 34.2 (32.3–36.5) | 767 (536–1149) | 600 (408–906) | 14.4 (9.75–21.7) | 36.1 (24.8–54.2) | 67.2 (65.4–69.6) | 57.0 (54.3–59.1) | 330 (225–523) | 258 (170–402) | 491 (324–793) | 66.4 (55.1–75.0) |
| **WHO-AFRO subregion** | | | | | | | | | | | | | | |
| Central Africa | 8.21 | 19.4 | 60.0 (56.0–63.7) | 46.8 (42.5–50.5) | 146 (99.6–221) | 127 (87.5–193) | 15.4 (10.7–23.4) | 48.9 (33.7–72.0) | 70.5 (67.4–72.9) | 61.2 (58.0–64.2) | 77.0 (51.4–124) | 47.6 (29.9–78.1) | 96.7 (60.9–155) | 67.0 (54.6–79.0) |
| East Africa | 12.0 | 24.4 | 41.0 (38.4–43.5) | 21.1 (18.8–23.7) | 203 (141–310) | 128 (85.8–192) | 10.7 (7.14–16.0) | 21.9 (15.0–31.6) | 63.2 (61.2–65.5) | 50.4 (47.0–53.7) | 56.8 (36.9–86.4) | 68.1 (46.7–109) | 128 (87.1–208) | 65.3 (56.7–72.4) |
| Southern Africa | 3.48 | 23.6 | 48.1 (46.2–50.4) | 34.1 (30.5–38.2) | 63.0 (44.5–94.7) | 51.4 (33.2–79.8) | 14.7 (9.52–22.9) | 35.2 (23.1–55.4) | 68.4 (66.1–71.2) | 56.5 (54.3–59.4) | 32.7 (22.4–51.4) | 17.5 (10.4–30.5) | 43.7 (29.1–69.2) | 71.9 (58.8–81.9) |
| West Africa | 16.3 | 23.0% | 56.6 (53.2–59.6) | 40.5 (35.7–44.3) | 342 (242–517) | 280 (197–425) | 17.2 (12.1–26.1) | 43.5 (31.2–65.3) | 69.1 (66.6–72.3) | 58.7 (55.2–61.4) | 161 (112–263) | 113 (74.5–178) | 225 (153–352) | 66.6 (54.1–77.4) |
| **HBHI versus non-HBHI countries** | | | | | | | | | | | | | | |
| Non-HBHI | 18.1 | 24.1 | 38.1 (36.4–40.6) | 25.1 (22.6–26.9) | 288 (197–436) | 215 (144–328) | 11.8 (7.93–18.1) | 26.3 (17.7–39.7) | 67.2 (65.4–69.6) | 53.8 (51.4–56.5) | 93.2 (63.3–145) | 116 (79.9–188) | 160 (104–251) | 58.3 (50.5–65.9) |
| HBHI | 23.7 | 21.6 | 60.0 (57.7–62.3) | 41.3 (38.7–44.5) | 482 (339–720) | 386 (270–590) | 16.3 (11.4–24.9) | 44.1 (30.9–66.1) | 70.1 (67.9–72.7) | 58.5 (55.5–61.3) | 234 (162–378) | 141 (91.2–234) | 331 (223–533) | 69.8 (57.9–81.2) |

Table summarizes the changes in the intrinsic burden of severe maternal anaemia (Hb < 7 g dl⁻¹) based on the effect of malaria on Hb evaluated at 200 days' gestation, caused by reductions in malaria transmission since 2000, and the effect of IPTp-SP at the current coverage levels, based on effect of IPTp on Hb evaluated during the third trimester. Values in parentheses denote 95% CrI. AFRO, World Health Organization African Region; HBHI, high burden to high impact. Central Africa: Angola, Cameroon, Central African Republic, Chad, DRC, Equatorial Guinea, Gabon, Republic of the Congo. East Africa: Burundi, Djibouti, Eritrea, Ethiopia, Kenya, Madagascar, Rwanda, Somalia, South Sudan, Tanzania, Uganda. Southern Africa: Botswana, Malawi, Mozambique, Namibia, South Africa, Zambia, Zimbabwe. West Africa: Benin, Burkina Faso, Côte d'Ivoire, The Gambia, Ghana, Guinea, Guinea-Bissau, Liberia, Mali, Niger, Nigeria, Senegal, Sierra Leone, Togo. African HBHI countries: Burkina Faso, Cameroon, DRC, Ghana, Mali, Mozambique, Niger, Nigeria, Tanzania, Uganda. Non-HBHI African countries with sustained malaria transmission: Angola, Botswana, Burundi, Central African Republic, Chad, Côte d'Ivoire, Djibouti, Eritrea, Ethiopia, Gabon, The Gambia, Guinea, Guinea-Bissau, Kenya, Liberia, Madagascar, Malawi, Mauritania, Namibia, Rwanda, Senegal, Sierra Leone, Somalia, South Africa, South Sudan, Sudan, Togo, Zambia, Zimbabwe.

coverage and preserving recent malaria control gains while pursuing further reductions.

We estimate that declines in the intrinsic risk of anaemia lagged reductions in malaria exposure, with an ~32% drop in exposure leading to only an ~22% decline in the risk of moderate-to-severe anaemia. This reflects a fundamental shift in the landscape of malaria in pregnancy burden. Primigravidae remain the most vulnerable to malaria-attributable anaemia, but as fewer women are now infected during their first pregnancy, multigravid women consequently have lower protective immunity. In a counterfactual scenario whereby the 2023 pregnancies experienced 2000 transmission levels, severe anaemia risk in primigravidae would have been sixfold higher than in multigravidae. Based on actual 2023 exposure levels, this gap narrowed to threefold, with the overall intrinsic burden in multigravidae effectively unchanged relative to that which would have occurred under the 2000 exposure levels.

This evolving immunological landscape underscores an unintended consequence of malaria control and leaves progress in malaria in pregnancy prevention precarious. The protective effects of prior malaria control have reduced primigravid exposure, but at the cost of reduced immunity in multigravidae. This means any disruption in IPTp provision is doubly concerning: it would not only allow the re-emergence of malaria-attributable anaemia in primigravidae, who remain highly susceptible, but would also expose multigravidae, who increasingly lack the immunological protection acquired through repeated infections, to greater risk. Should IPTp coverage decline in parallel with resurging transmission, the result could be a broader resurgence in severe maternal anaemia across all gravidities. These findings reinforce the critical need to safeguard IPTp-SP as an essential intervention to prevent malaria-specific burden and avoid compounding strain on maternal health services, which already face uncertain resourcing in future years[41].

Our modelled risk trajectories also highlight a critical gap in current ANC policy—that is, the first trimester when sulfadoxine-pyrimethamine is contraindicated because of concerns about potential teratogenicity[10,42]. We find that the risk of malaria-attributable anaemia rises steeply throughout gestation, particularly in first-time pregnancies. However, while early ANC attendance in the first trimester is promoted to improve maternal health outcomes, current WHO guidance, given the contraindication of sulfadoxine-pyrimethamine, lacks clarity and operational feasibility, with suggestions to schedule an ANC visit at the 13th week of gestation solely for the provision of sulfadoxine-pyrimethamine or to dispense the medication in the first trimester with instructions for delayed self-administration[43]. First-trimester infection is associated with a higher risk of anaemia even when IPTp is administered at subsequent visits[44]. In the absence of a safe first-trimester preventive strategy, programmes could consider systematic screening and treatment of malaria infection at early ANC visits, particularly among primigravidae, where infections are more severe and more likely to be detected.

Our analysis does not account for a range of factors that could introduce heterogeneity in the effectiveness of IPTp towards ameliorating malaria-associated anaemia burden. In particular, our analysis does not explore potential changes in IPTp effectiveness because of evolving sulfadoxine-pyrimethamine resistance since the initial trials. The extent to which such resistance undermines the protective effect of IPTp-SP on anaemia remains under scientific debate[45], further complicated by the potential effects of sulfadoxine-pyrimethamine on non-malarial causes of maternal and neonatal morbidity[46], which our study does not attempt to quantify. However, no alternative intervention has demonstrated consistently superior performance in preventing malaria in pregnancy. Any reduction in efficacy due to resistance does not diminish the need for continued IPTp delivery; on the contrary, it heightens the risk posed by any future resurgence in malaria transmission, potentially compounding maternal and neonatal anaemia-related burden. We also do not incorporate other causes of Hb variation, such as nutritional deficiencies[47], HIV[48] or helminth

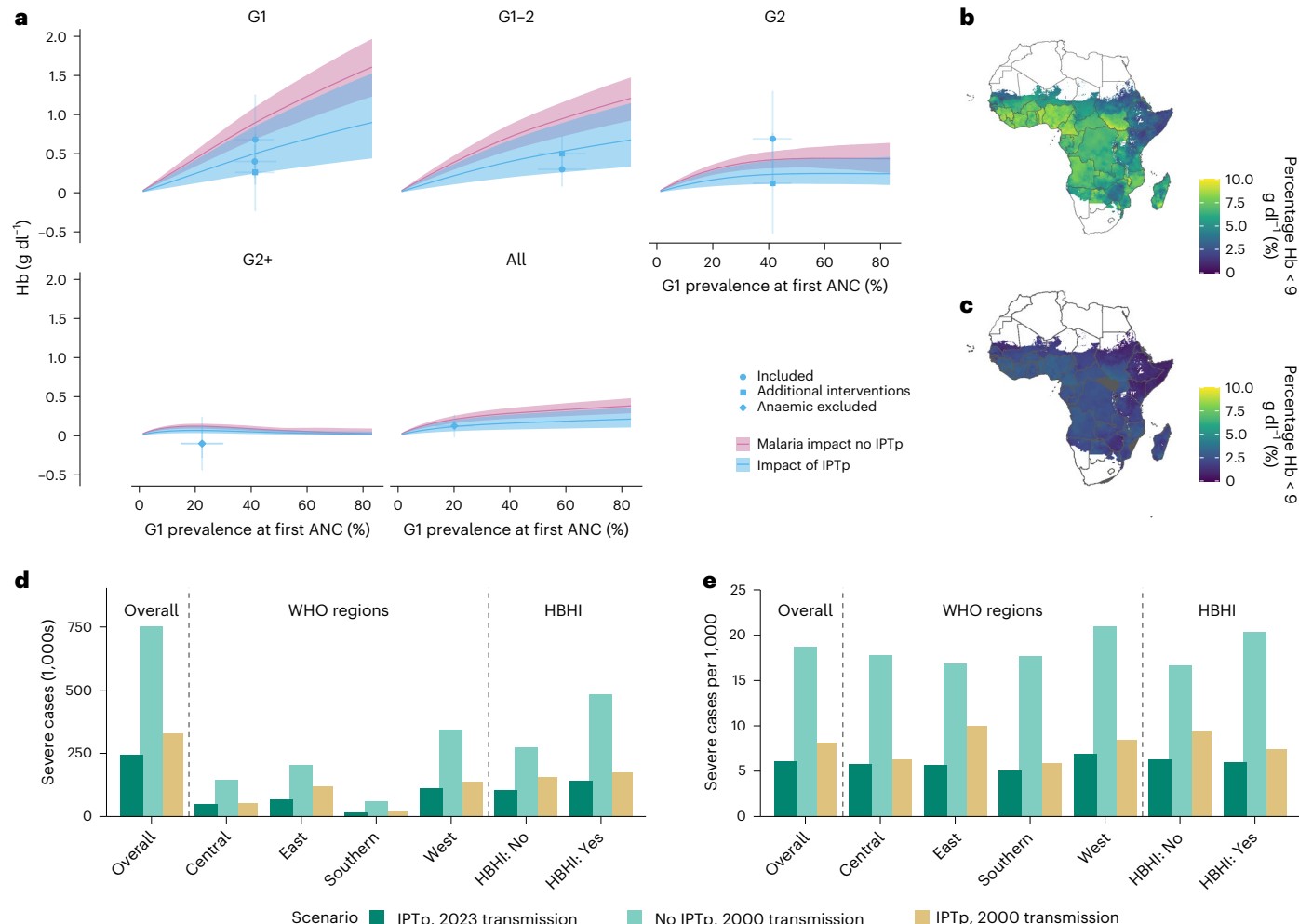

**Fig. 5 | Estimating the extent to which IPTp-SP mitigates the effect of malaria on Hb during pregnancy. a**, Model fit to third-trimester Hb differences observed in IPTp-SP trials. The shapes show trial-specific estimates of the absolute effect of IPTp-SP on Hb concentration. The circles represent trials included in the model fitting[33–35]; the square represents a trial involving monthly sulfadoxine-pyrimethamine (SP) administration[33]; the diamond represents a trial that coadministered ITNs alongside placebo or SP[35]; and the triangle represents a trial in which anaemic women were excluded at enrolment (Hb < 7 g dl⁻¹)[28,36]. The vertical error bars indicate the 95% Wald CIs, computed using a Welch-type standard error allowing for different sample sizes and variances in comparator groups, observed in each trial; the horizontal bars represent the 95% posterior estimates of primigravid PCR prevalence. The purple ribbons show the estimated

mean absolute reductions in Hb attributable to malaria in the control arms; the blue ribbons show the corresponding mitigated estimates assuming the IPTp-SP efficacy inferred from model fitting. **b**, Map of the modelled proportion of all pregnant women with moderate-to-severe anaemia (Hb < 9 g dl⁻¹) by the beginning of the third trimester (200 days' gestation) in the absence of IPTp-SP. **c**, Equivalent map assuming that IPTp-SP provides the level of protection estimated from fitting the model as presented in **a**. **d**, Estimated number of severe maternal anaemia cases given the current IPTp coverage and transmission levels (2023), and under historical counterfactuals assuming no IPTp or no decline in malaria transmission. Results are shown separately for WHO regions and HBHI strata. **e**, Equivalent to **d** but presented as the burden per 1,000 pregnancies. Administrative boundaries were obtained from GADM v.3.6.

infections[49], probably contributing to heterogeneity in anaemia burden. However, our ability to reproduce Hb trajectories across trials suggests that malaria is a key driver in these settings and that our estimates reflect the scale of effect of malaria on maternal anaemia. We also rely on model-based inference of prior pregnancy exposure from contemporary primigravid prevalence—a limitation shared by all attempts to characterize pregnancy-specific immunity in the absence of lifelong data. Meanwhile, our findings that the effect of malaria in primigravid women did not vary notably across study sites and that pregnancy-specific immunity acquired from exposure during previous pregnancies, rather than that acquired by exposure to malaria before the first pregnancy, is the prime modifier of malaria impact on Hb during pregnancy. However, this does not preclude a much more prominent role of non-pregnancy-acquired immunity in lower transmission settings than assessed in this study (that is, <10% PCR prevalence). Given that these are settings where exposure is

increasingly rare, such effects may not substantially alter our overall burden estimates but may have important consequences for individuals affected and highlight the importance of maintaining protection through accurate diagnosis of infection early in pregnancy and the exercise of caution when determining transmission thresholds below which IPTp should not be considered.

Future work will integrate these refined estimates of malaria-related anaemia into broader assessments of malaria in pregnancy burden, including effects on neonatal outcomes such as LBW and early mortality. As funding constraints loom over global malaria and maternal health programmes, it is essential that malaria in pregnancy receives adequate attention in prioritization frameworks. These findings strengthen the case for safeguarding IPTp-SP coverage and other measures targeted towards protecting women from malaria as both an intervention and a cornerstone of maternal and neonatal health.

## Methods

This section provides a detailed account of the data, modelling framework, and analytical procedures used in our study. We begin by describing the source studies and inclusion criteria for the individual-level data. We then outline the inferential framework, including how we adjusted for anaemia-related exclusion bias (full code for which is available at https://patrickgtwalker.github.io/Anaemia_malaria_fitting_open/vignette/malaria_hb_vignette.html). This is followed by descriptions of the Hb dynamics model, the model of malaria infection in pregnancy (full code for which is available https://github.com/patrickgtwalker/malaria_in_pregnancy_istp_model_open/releases/tag/v1.0-istp_paper) and their integration. We conclude with the estimation of IPTp-SP effectiveness and the procedure for generating continent-wide estimates of malaria-attributable anaemia burden and intervention impact.

### Malaria and anaemia data

For our analysis, we included studies that fulfilled the following criteria: (1) women received no recorded intermittent preventive treatment during pregnancy at the time of sampling; (2) *P. falciparum* infection status according to PCR and Hb concentration were recorded at the same point during gestation; (3) women were recruited throughout the first and second trimester.

We analysed data from four trials that compared intermittent preventive treatment in pregnancy with IPTp-SP to alternative strategies, including IPTp with dihydroartemisinin–piperaquine, with or without azithromycin, or intermittent screening and treatment (IST) with dihydroartemisinin–piperaquine or artemether–lumefantrine. These were a trial conducted in southern Malawi between 2011 and 2013 (Malawi IST)[24], a trial in western Kenya between 2012 and 2014 (Kenya IST)[25], a multi-country trial conducted in Burkina Faso, Mali, The Gambia and Ghana between 2010 and 2011 (West Africa trial)[27], and the IMPROVE-1 trial conducted between 2018 and 2019 across Kenya, Malawi and Tanzania (IMPROVE-1)[26]. In all four trials, women had not yet received IPTp at the time of enrolment.

Most trials recruited women regardless of Hb concentration and gravidity, with Hb measured with a point-of-care haemoglobinometer using capillary blood samples. However, two trials—Malawi IST and Kenya IST—excluded women with severe anaemia (Hb < 7 g dl⁻¹) while the West Africa trial recruited only primigravidae and secundigravidae (G1 or G2). Across the four studies, individual-level data were available for 1,803 women from the Malawi IST, 1,396 from the Kenya IST, 5,208 from the West Africa trial (including 1,413 from Burkina Faso, 1,193 from The Gambia, 1,298 from Ghana and 1,304 from Mali) and 4,201 from IMPROVE-1 (1,297 from Kenya, 1,283 from Malawi and 1,621 from Tanzania).

In total, we analysed data from 12,608 women across 7 countries—Burkina Faso, The Gambia, Ghana, Kenya, Malawi, Mali and Tanzania. These included individual-level records on Hb concentration, PCR-based malaria status, gravidity and gestational age at enrolment. We used gestational age at enrolment as a proxy for the length of time a malaria infection remained untreated, allowing us to characterize Hb dynamics across the first and second trimesters of pregnancy.

### Disentangling age and gravidity

Given the close correlation between maternal age and gravidity, we first fitted mixed-effects regressions to assess whether each independently modified the effect of malaria on Hb. We adjusted for gestational age using a natural cubic spline and included a random intercept for study site to account for between-site heterogeneity. Three nested specifications were evaluated: (1) gravidity-specific malaria effects; (2) gravidity-specific effects plus a global interaction between malaria and age; and (3) a three-way interaction between malaria, age and gravidity. Model selection was guided by the Akaike information criterion; and the strength of evidence for improved fit attributable to age was assessed

using likelihood ratio tests. To visualize patterns transparently, we also ran stratum-specific contrasts using estimated marginal means. Finally, we compared gravidity-specific coefficients from models with and without adjustment for age, confirming that the simpler specification adopted in our inferential framework (malaria according to gravidity interaction only) did not materially alter gravidity-specific results.

**Inferential framework.** We developed a hierarchical Bayesian model to characterize Hb concentrations across pregnancy, accounting for malaria infection, gestational age, gravidity and exposure history. Our inferential framework was structured to handle two key scenarios: studies with unrestricted enrolment and studies where women with severe anaemia (Hb < 7 g dl⁻¹) were excluded at baseline.

**Studies where women were recruited at enrolment without anaemia-specific exclusion criteria.** For studies without anaemia-based exclusion, we modelled the joint likelihood of observed Hb concentrations given malaria status (as measured using PCR), gestational age and gravidity. Let $H_s = \{h_{s,i}; i \in 1..n_s\}$ denote the set of Hb values for study $s$, with $M_s$, $T_s$ and $G_s$ representing malaria infection status, gestational age and gravidity, respectively. The parameters $\theta$ define the Hb trajectory model and $\Theta$ represents parameters from our existing malaria-in-pregnancy transmission model.

We specified the posterior as:

$$P(\theta|H_s, M_s, T_s, G_s \Theta) \propto P(H_s|M_s, T_s, G_s, \Theta, \theta)P(\theta) \qquad (1)$$

To account for prior exposure, we incorporated an integrated likelihood:

$$P(H_s|M_s, T_s, G_s, y_{sg}, n_{sg}, \Theta, \theta, \pi_{s1}) = \prod_{i=0}^{n_s} f(h_{si}, m_{si}, t_{si}, g_{si}, \Theta, \theta, \pi_{s1}) \qquad (2)$$

where

$$\begin{aligned} &f(h_{si}, m_{si}, t_{si}, g_{si}, \Theta, \theta, \pi_{s1}) \\ &= \int P(h_{si}|m_{si}, t_{si}, g_{si}, e_{si}, \theta)P(e_{si}|\pi_{s1}, g_{si}, \Theta)de_{si}P(\pi_{s1}|y_{s1}, n_{s1}). \end{aligned} \qquad (3)$$

$P(h_{si}|m_{si}, t_{si}, g_{si}, e_{si}, \theta)$ represents the model linking malaria exposure in previous pregnancies and other model parameters to be estimated to Hb concentration at enrolment (see 'Hb dynamics model'). $P(e_{si}|\pi_{s1}, g_{si}, \Theta)$ represents the exposure to malaria in previous pregnancies given primigravid prevalence within a setting and individual-level gravidity, uncertainty in which is integrated over (see 'Malaria in pregnancy model'). $P(\pi_{s1}|y_{s1}, n_{s1})$ represents a binomially distributed likelihood of primigravid prevalence given the observed infection status of primigravids within the study. This accounts for uncertainty in prior malaria exposure $e_{si}$ as a function of observed primigravid PCR prevalence $\pi_{s1}$ (see 'Malaria in pregnancy model'). $P(\pi_{s1}|y_{s1}, n_{s1})$ represents a binomially distributed likelihood of primigravid prevalence given the observed infection status of primigravids within the study.

**Studies with anaemia-based exclusion.** To adjust for exclusion of women with Hb < 7 g dl⁻¹, we extended the model to include latent censoring and marginalized over the unobserved distribution of excluded women. Let $Z$ be the Hb threshold, and $c_s$ the number of excluded women. We introduced the additional parameters $\Pi = \{\pi_{sg}; g = 1 \dots n_g\}$, the set of prevalences according to gravidity category before censoring (modelled as log-odds $o_{sg}$), $\Omega = \{\sigma_{sg}; g = 1 \dots n_g\}$; the proportion of women in each gravidity category before censoring (modelled as multipliers relative to the proportion of primigravid women $RR_{sg}$) and $\Lambda_s$ represent the set of parameters consisting of the mean and s.d. of the set of gestational age at enrolment before censoring (modelled according to a Beta distribution scaled between an assumed minimum and maximum enrolment time of 80 and 200 and discretized into integer days).

The adjusted posterior becomes:

$$P(\theta, \Omega_s, \Lambda_s | H_s, M_s, T_s, G_s, c_s, Z, \Theta)$$

$$\propto P^*(H_s | M_s, T_s, G_s, Z, \Theta, \theta) P(c_s | p_c, n_s)$$

$$\times P(p_c | Z, \theta, \Theta, \Pi_s, \Omega_s, \Lambda_s) P(M_s, T_s, G_s | Z, \theta, \Theta, \Pi_s, \Omega_s, \Lambda_s) P(\theta, \Pi_s, \Omega_s, \Lambda_s) \quad (4)$$

where $P^*(.)$ denotes likelihoods conditioned on non-censoring:

$$P^*(h_{si}|\dots) \propto \frac{f(h_{si}, m_{si}, t_{si}, g_{si}, \Theta, \theta, \pi_{s1})}{\int_{h=0}^{Z} f(h, m_{si}, t_{si}, g_{si}, \Theta, \theta, \pi_{s1}) \, dh} \quad (5)$$

with $f(h, m_{si}, t_{si}, g_{si}, \Theta, \theta, \pi_{s1})$ representing the uncensored likelihood from equation (3).

The probability of exclusion is:

$$P(c_s | p_c, n_s) \propto p_c^{c_s} (1 - p_c)^{n_s} \quad (6)$$

with $p_c$ marginalized over malaria status, gravidity and gestational age.

This framework allows us to recover the likely distribution of Hb in the underlying population, appropriately corrected for bias introduced by the exclusion criteria.

## Hb dynamics model

We evaluated a range of models to describe Hb concentration across pregnancy, incorporating different assumptions regarding the effect of malaria infection, gravidity and gestational timing. Models were compared using the deviance information criterion, and the final model was selected based on goodness of fit and biological plausibility.

We modelled individual-level Hb values as following a normal distribution:

$$h_{si} \sim N(\mu_{si}, \sigma) \quad (7)$$

where $\mu_{si}$ is the expected Hb for individual in study $s$, and $\sigma$ is the s.d. assumed constant across individuals. The mean $\mu_{si}$ is defined as:

$$\mu_{si} = \alpha(t_{si}) + m_{si}\beta(t_{si})\phi(e_{si}) + \gamma_{g_{si}} + \delta_s \quad (8)$$

In this formulation, $\alpha(t)$ is a cubic spline function representing the typical trajectory of Hb across gestation in uninfected primigravidae. The function $\beta(t)$ captures the time-varying effect of malaria infection on Hb. The binary variable $m_{si}$ indicates whether individual was infected with malaria at enrolment. The effect of prior malaria exposure is incorporated via the function $\phi(e)$, which modifies the effect of malaria infection as a function of $e_{si}$, the number of prior pregnancies in which the woman was likely exposed to malaria. The term $\gamma_{g_{si}}$ represents a gravidity-specific adjustment to the Hb intercept, with $\gamma_1 = 0$ for primigravidae. A study-specific intercept $\delta_s$ accounts for site-level differences in Hb measurements.

The exposure adjustment function $\phi(e)$ was defined as a power-law function:

$$\phi(e) = \left(1 + (e/v)^\kappa\right) \quad (9)$$

where $v$ and $\kappa$ are the shape parameters estimated from the data.

We also considered alternative models, including those with no effect of malaria ($\beta(t) = 0$), a constant malaria effect ($\beta(t) = \beta$), gravidity-specific malaria susceptibility independent of exposure history ($\phi(g) = v_g$) and models with or without gravidity-dependent baseline Hb (that is, $\gamma_g = 0$) (see Supplementary Table 1 for a full list of models tested and Supplementary Table 2 for a list of parameter definitions, prior distributions and posterior summaries for the final selected model).

## Malaria in pregnancy model

Full details of our model of malaria in pregnancy are available elsewhere[23] and our code has been placed in an open-source repository (https://github.com/patrickgtwalker/malaria_in_pregnancy_istp_model_open). In this section, we detail key features and our approach to capturing patterns of previous exposure according to site and gravidity within our fitting.

The model follows a cohort of women of child-bearing age from 15 to 49 years old. Age at each conception throughout a woman's lifetime, denoted by $C = \{C_g\}$, where $g = 1, .., F$ and $F$ represents the total pregnancies, generated according to gravidity-specific fertility rates stratified according to 5-year age groups (typically calculated from Demographic Health Surveys (DHS) and MIS[50]).

The extent to which women are exposed to malaria in pregnancy is generated using a customized version of an established malaria transmission model (Supplementary Fig. 1). At conception, women are assigned an infection state: uninfected (S), untreated clinical malaria (D), treated clinical malaria (T), prophylaxis after treatment (P), asymptomatic patent (A) or sub-patent (U). The probabilities of initial states depend on the entomological inoculation rate, age and individual heterogeneity in mosquito exposure (modelled via a Gauss–Hermite approximation to a log-normal distribution).

Each woman is assigned a force of infection reflecting her expected rate of exposure throughout gestation. Times of blood-stage infections during pregnancy, $B_g = \{B_{g,i}\}$, are generated as follows:

- If the woman is infected at conception, $B_{g,1} = 0$, else $B_{g,1} = X(\lambda)$, where $X(.) \sim \text{Exp}(.)$
- Subsequent infections occur as $B_{g,j+1} = B_{g,j} + X(\lambda)$

Clearance times for each infection (assuming no pregnancy-specific effects), $K_g = \{K_{g,i}, i = 1..n_B\}$, are similarly drawn according to the underlying transmission model parameters.

From week 13 of gestation onwards, peripheral infections may sequester in the placenta. Sequestered infections $P_g = \{P_{g,i}, i = 1 \dots n_P\}$ resolve at times $R_g = \{R_{g,i}, i = 1 \dots n_P\}$ with durations depending on gravidity-specific pregnancy immunity.

PCR positivity at time $t$ in pregnancy, $x_g^P(t)$, is defined as

$$x_g^P(t) = 1\left(\left[\sum_{j=1}^{n_B}\left[1\left(B_{g,j} < t, K_{g,j} > t\right)\right] + \sum_{j=1}^{n_P}\left[1\left(P_{g,i} < t, R_{g,i} > t\right)\right]\right] \geq 1\right) \quad (10)$$

We then link this model to our Hb model via:

$$h_g(t) \sim N(\mu_g(t), \sigma), \quad \mu_g(t) = \alpha(t) + x_g^P(t)\beta(t)\phi(e_g) + \gamma_g \quad (11)$$

where $\mu_g(t)$ is the expected Hb level given gestational age $t$, gravidity $g$, infection status $x_g^P(t)$ and $e_g$ the number of prior pregnancies exposed to malaria.

Simulation outputs from this model are used both to predict Hb distributions under different exposure scenarios (see 'Generating estimates of the burden of malaria-associated anaemia and impact of IPTp across Africa') and to estimate the expected distribution of prior exposure given infection, which is used within the likelihood function during model fitting (see 'Inferential framework').

## Fitting the model to the data

We fitted the model using the drjacoby R package (v.1.5.4), which provides a flexible platform for Bayesian inference via Markov Chain Monte Carlo. We used weakly informative priors (Supplementary Table 1) and ran two chains for each model to monitor convergence.

Chains were run with sufficient burn-in and thinning to achieve an effective sample size of at least 1,000 per parameter. Convergence was assessed via trace plots, Gelman–Rubin diagnostics and posterior density comparisons between chains.

All posterior summaries and parameter estimates are based on the retained post-burn-in samples. Details of the priors, hyperparameters and model variants tested are summarized in Supplementary Tables 1 and 2.

### Estimating the effectiveness of IPTp-SP upon Hb levels

To estimate the effect of IPTp-SP on maternal Hb, we analysed studies from a Cochrane review of the effect of the intervention versus placebo. Data from three studies were included in our primary analysis on the basis that they reported the mean and variance of Hb levels in both trial arms and did not exclude women based on Hb concentration, with additional studies that had such exclusion criteria included for qualitative comparison (Supplementary Table 4).

We defined the proportional effectiveness of IPTp-SP as the fraction according to which the intervention mitigates the Hb reduction attributable to malaria. Let $Y = \{y_r\}$ and $S = \{s_r\}$ denote the mean and s.d. of Hb increases observed in each trial and gravidity stratum pairing. Let $M = \{m_r\}$ be the modelled malaria-attributable reduction in Hb in the absence of IPTp-SP. We defined the likelihood of the observed data under this model as:

$$l(Y, S|\varepsilon, M) = \prod_{r=1}^{n_r} l(y_r, s_r|\varepsilon, m_r), \text{ where } y_r \sim N(\varepsilon \times m_r, s_r) \quad (12)$$

To generate the predicted $m_r$ values, we used our transmission and Hb models, conditioning on primigravid PCR prevalence, gravidity distribution and timing of malaria exposure. As the studies included only reported slide-positive prevalence, we used previously estimated relationships between microscopy and PCR detection to infer the PCR prevalence in each trial[23]. Specifically, we assumed that the microscopy prevalence in primigravidae followed a Beta distribution, reflecting the posterior uncertainty under a binomial model with a flat prior:

$$\rho_r^m \sim \text{Beta}(sm_{1r} + 1, n_{1r} - sm_{1r} + 1) \quad (13)$$

where $sm_{1r}$ is the number of positive primigravidae according to slide microscopy and $n_{1r}$ the total sampled.

We then sampled 1,000 realizations of PCR prevalence using a logistic transformation with uncertainty propagated from previously estimated parameters:

$$\rho_{ri}^p = \text{expit}\left(\frac{\text{logit}(\rho_{ri}^{sm}) - \zeta_i}{1 + \eta_i}\right) \quad (14)$$

where $\text{expit}(x) = 1/(1 + e^{-x})$. For studies lacking gravidity-specific microscopy prevalence (for example, ref. 37), we used study-level information and simulated the odds ratio between primigravidae and secundigravidae using a triangular distribution, incorporating empirical bounds from the ISTp trials.

$\rho_{ri}^p$ for the $i$th draw was then used to simulate the expected malaria-attributable reduction associated with that draw $m_{ri}$ and the likelihood was integrated over the 1,000 simulations. We maximized the marginal likelihood with respect to $\varepsilon$ and used importance sampling to generate a posterior distribution.

Extended Data Fig. 1 shows the model fitted to the Hb data from each study. Results were consistent with those from other trial arms not included in the primary fitting, for example, monthly sulfadoxine-pyrimethamine regimens and high-gravidity strata in high-transmission settings supporting the robustness of the estimate.

### Generating estimates of the burden of malaria-associated anaemia and impact of IPTp across Africa

To estimate the continent-wide burden of malaria-attributable anaemia in pregnancy, we simulated Hb trajectories under observed transmission conditions in 2023 using our combined model of malaria in pregnancy and Hb dynamics. The model was implemented across a 0.2-degree (approximately 5 km²) spatial grid, aligned with the MAP 2023 posterior distribution of slide prevalence in children aged 2–10 years. For each location, 100 realizations of slide prevalence were drawn to reflect spatial and epidemiological uncertainty.

Fertility inputs were derived from the most recent available DHS or MIS surveys for each country. Where survey data were not available, we used data from a demographically similar neighbouring country matched by TFR. Fertility rates were stratified according to 5-year maternal age groups and gravidity and disaggregated according to urban or rural setting. Urban and rural classification was determined at 1-km² resolution using WorldPop 2020 population density maps, applying a threshold of 386 persons per km² to define urban areas. This classification was then aggregated to the 5-km² model resolution.

We used these inputs to simulate a synthetic cohort of pregnancies according to country, gravidity and location, calculating the expected distribution of Hb concentration under observed 2023 malaria transmission conditions. Estimates of moderate-to-severe anaemia prevalence, as well as the burden specifically attributable to malaria infection, were aggregated across all grid cells, weighted by the local fertility rate and population size, and then scaled to match WHO-estimated country-level pregnancy counts. These counts were based on United Nations-estimated births, adjusted for pregnancy losses (miscarriage, stillbirth and abortion) using standard multipliers.

To quantify the effect of declining malaria transmission on anaemia burden, we re-ran these simulations using the 2000 transmission levels (that is, pre-decline), drawing from the MAP 2000 posterior distribution of slide prevalence in children. This enabled a counterfactual comparison in which the same underlying population structure was retained but malaria exposure was elevated to historical levels. Comparing model outputs under the 2000 versus 2023 transmission scenarios provided estimates of the number of moderate-to-severe anaemia cases averted because of transmission reductions over the past two decades.

Finally, we incorporated the posterior distribution of IPTp-SP effectiveness derived from trial data (see above), applying the estimated mitigation fraction ($\varepsilon$) to malaria-attributable reductions in Hb. This allowed us to estimate the number of moderate-to-severe anaemia cases that would be averted under full IPTp-SP coverage. We then simulated the real-life impact of IPTp using WHO estimates of the coverage of these estimates (except for Kenya and Zimbabwe, which implements IPTp sub-nationally where we obtained coverage for relevant regions from the latest population-based surveys[51,52]) assuming a multiplicative per-dose effect.

### Ethics and inclusion statement

This work builds on long-standing collaborations between modellers, clinical trialists, epidemiologists and malaria surveillance experts spanning more than a decade and involving institutions based in both malaria-endemic countries and high-income research settings. The research questions addressed in this study, including the need for robust estimates of malaria-attributable maternal anaemia and the impact of preventive interventions, were defined through these collaborations well before the present analysis and reflect priorities identified by local investigators and malaria control stakeholders. The individual-level data used in this study were derived from multi-country clinical trials and surveillance studies conducted in malaria-endemic settings, led or co-led by investigators based in those settings. Local researchers were involved in study design, data collection, data interpretation and authorship of the original studies, with data governance arrangements established at the time of data collection. Authorship in this manuscript reflects substantive intellectual contributions consistent with Nature Portfolio criteria. All primary studies contributing data received approval from relevant local and national ethics review committees and institutional review boards, as reported in the original publications. The present work is a secondary analysis

of de-identified data and did not require additional ethical approval. Results are presented as aggregated and modelled estimates and do not pose risks of stigmatization or harm to participants.

## Reporting summary

Further information on research design is available in the Nature Portfolio Reporting Summary linked to this article.

## Data availability

We refer to studies by their names as listed in Supplementary Table 1: The West Africa IST dataset[27] is archived at the LSHTM Data Compass repository (https://doi.org/10.17037/DATA.2) and is available under controlled access upon request. The Malawi IST dataset[24] is available via the WorldWide Antimalarial Resistance Network (WWARN), with access subject to review by the WWARN Data Access Committee. The Kenya IST[25] and IMPROVE-1[26] datasets are available from the study investigators upon reasonable request and are subject to appropriate data use agreements and ethical approvals. Malaria prevalence surfaces (MAP), demographic and fertility data (DHS/MIS) and population estimates (WorldPop) are publicly available and cited in the paper.

## Code availability

All code used for this analysis is open access and available via the following repositories: Hb dynamics and anaemia risk model (https://github.com/patrickgtwalker/Anaemia_malaria_fitting_open); and integrated malaria in pregnancy transmission model (https://github.com/patrickgtwalker/malaria_in_pregnancy_istp_model_open). Both repositories include full documentation and implementation guidance. Feedback and contributions are welcome.

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

## Acknowledgements

We thank all participants and data collectors involved in the original studies providing data for this study. S.I.L. and P.G.T.W. acknowledge funding from the Gates Foundation (INV-005289); S.I.L., P.G.T.W., R.V. and M.C. recognize additional support provided by the UK Medical Research Council (MRC) through the MRC Centre for Global Infectious Disease Analysis (grant reference MR/R015600/1), jointly funded by the MRC, the European Union's EDCTP2 programme and Community Jameel. M.C. is supported by a Sir Henry Dale Fellowship jointly funded by the Wellcome Trust and the Royal Society (grant no. 220658/Z/20/Z). A.N. acknowledges funding from the Gates Foundation (INV-061337). The findings and conclusions in this report are those of the authors and do not necessarily represent the official position of the Centers for Disease Control and Prevention.

## Author contributions

S.I.L., P.G.T.W. and R.V. conceived and designed the study. S.I.L. and P.G.T.W. conducted the analysis and prepared the figures. J.R.G., D.J.W., M.C., J.E.W., C.K., K.B., J.D., A.M.N., S.M.T., K.O., S.O.C., K.K., S.K., B.G., M.D., D.C., H.T., F.O.t.K. and M.M. contributed data or aided with the interpretation of the data. All authors contributed to drafting and revising the manuscript.

## Competing interests

The authors declare no competing interests.

## Additional information

**Correspondence and requests for materials** should be addressed to Patrick G. T. Walker.

Sequoia I. Leuba[1,2], Robert Verity [1], Julie R. Gutman [3], Daniel J. Weiss [4], Matt Cairns [1], John E. Williams[5], Carole Khairallah[6], Kalifa Bojang[7], James Dodd[6], Abdisalan M. Noor[8], Steve M. Taylor [9], Kephas Otieno[10], Sheick Oumar Coulibaly[11], Kassoum Kayentao[12], Simon Kariuki[10], Brian Greenwood [2], Meghna Desai[3], Daniel Chandramohan[2], Harry Tagbor[13], Feiko O. ter Kuile [6], Mwayiwawo Madanitsa[14] & Patrick G. T. Walker [1] ✉

[1]MRC Centre for Global Infectious Disease Analysis, School of Public Health, Imperial College London, London, UK. [2]London School of Hygiene and Tropical Medicine, London, UK. [3]Malaria Branch, Division of Parasitic Diseases and Malaria, National Center for Emerging and Zoonotic Infectious Diseases, Centers for Disease Control and Prevention, Atlanta, GA, USA. [4]Malaria Atlas Project, Telethon Kids Institute, Perth Children's Hospital, Perth, Western Australia, Australia. [5]Dodowa Health Research Centre, Dodowa, Ghana. [6]Department of Clinical Sciences, Liverpool School of Tropical Medicine, Liverpool, UK. [7]MRC Unit, The Gambia at London School of Hygiene and Tropical Medicine, Fajara, The Gambia. [8]Harvard T.H. Chan School of Public Health, Boston, MA, USA. [9]Division of Infectious Diseases, Duke University School of Medicine, Duke University, Durham, NC, USA. [10]Kenya Medical Research Institute, Centre for Global Health Research, Kisumu, Kenya. [11]Faculty of Health Sciences, University of Ouagadougou, Ouagadougou, Burkina Faso. [12]Parasites and Microbes Research and Training Center, Mali International Center for Excellence in Research, University of Sciences, Techniques and Technologies of Bamako, Bamako, Mali. [13]University of Health and Allied Sciences, Ho, Ghana. [14]Department of Clinical Sciences, Malawi University of Science and Technology, Zomba, Malawi. ✉e-mail: patrick.walker@imperial.ac.uk

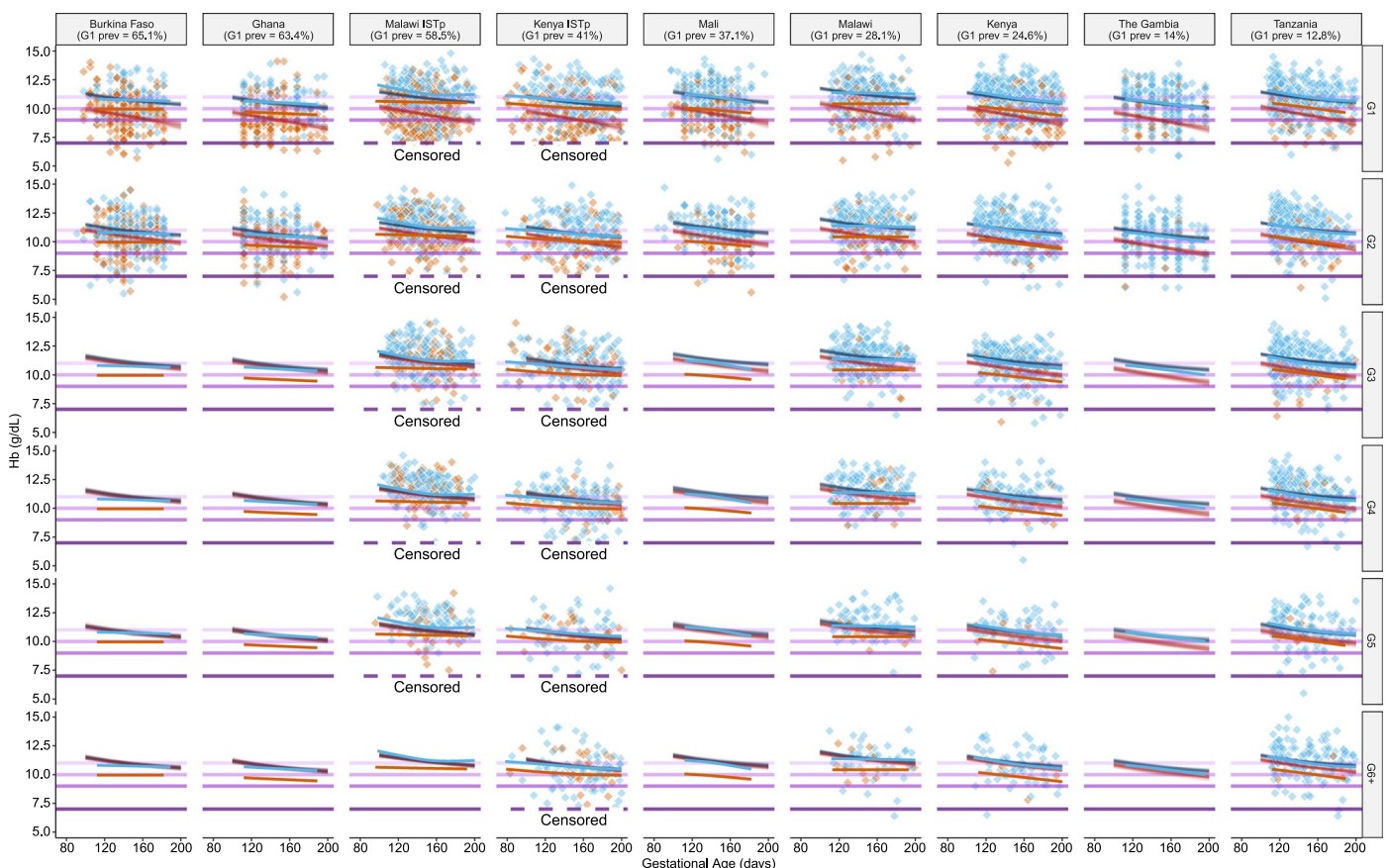

**Extended Data Fig. 1 | Comparison of individual-level hemoglobin concentrations and final model fit, stratified by trial site, gestational age, and gravidity.** Circles represent individual Hb (Hb) concentrations for participants who were PCR-positive (orange) or PCR-negative (light blue) at enrollment, plotted by gestational age. Locally Estimated Scatterplot Smoothing (LOESS) curves are shown in matching colors where data are sufficient; these are included for visual reference only. Red lines (for infected) and dark blue lines (for uninfected) show 1,000 draws from the joint posterior distribution of the final fitted model. Columns represent trial sites (ordered by primigravid PCR prevalence), and rows represent gravidity groups (with the bottom row labeled "6" including gravidity 6 and above). Thresholds of 11, 10, 9, and 7 g/dL are indicated by horizontal purple lines, with increasing darkness denoting lower values.

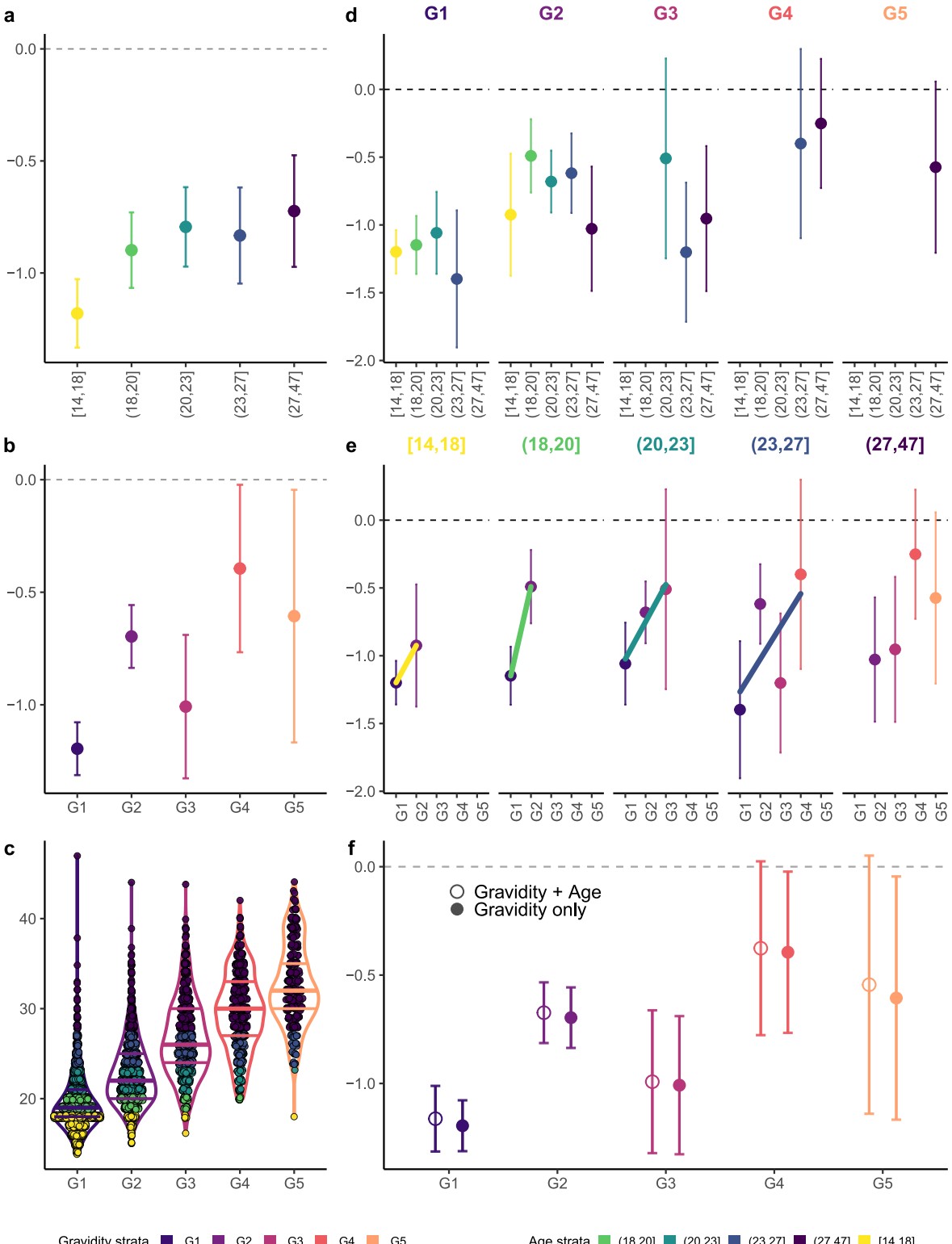

**Extended Data Fig. 2 | Age and gravidity in relation to malaria-associated hemoglobin reduction.** (**a**) Estimated change in hemoglobin due to malaria, stratified by maternal age quintile. (**b**) Estimated change in hemoglobin due to malaria, stratified by gravidity. (**c**) Correlation between maternal age and gravidity, shown using violin outlines (colored by gravidity) with overlaid individual observations (colored by age quintile), and median and interquartile range bars clipped to the violin edges. (**d**) Gravidity-specific trends in malaria-associated hemoglobin change across age quintiles. Trend lines are shown only where the interaction between malaria and age was significant within that gravidity stratum. (**e**) Age-specific trends in malaria-associated hemoglobin change across gravidities. Trend lines are shown only where the interaction between malaria and gravidity was significant at α = 0.05 within that age stratum and strata with fewer than 50 observations across malaria arms are excluded. (**f**) Comparison of gravidity-specific malaria effects estimated from models with and without adjustment for maternal age. Points show estimates with 95% Wald confidence intervals from mixed-effects models adjusting for gestational age and study site.

**Extended Data Table 1 | Summary of changes in the intrinsic burden[¥] of severe or moderate-to-severe maternal anaemia (Hb < 9 g/dL) due to reductions in malaria transmission since 2000, and the impact of IPTp-SP[±] at current coverage level.**

| Area | Pregnancies at risk | | % pregnancies exposed | | Intrinsic burden (000s) | | Intrinsic burden (per-1000 pregnancies) | | % burden in primigravidae | | IPTp Impact (000s) | | Burden Averted | |
|---|---|---|---|---|---|---|---|---|---|---|---|---|---|---|
| | Total (m) | G1 % | 2000 | 2023 | 2000 | 2023 | Overall | G1 | 2000 | 2023 | Averted | Remaining | Total | % |
| All Countries | 41.8 | 22.6% | 50.3% (49.1,52.0) | 34.2% (32.3,36.5) | 3039(2515,3853) | 2415(1977,3036) | 57.7 (47.3,72.6) | 125 (103,154) | 58.8% (55.8,61.6) | 49.2% (46.7,51.4) | 1102 (719,1611) | 1313 (850,1818) | 1727(1251,2425) | 57.2% (43.5,72.7) |
| **WHO-AFRO subregion** | | | | | | | | | | | | | | |
| Central Africa | 8.21 | 19.4% | 60.0% (56.0,63.7) | 46.8% (42.5,50.5) | 577 (468,726) | 517 (425,643) | 62.9 (51.8,78.2) | 174 (141,207) | 61.4% (57.6,65.0) | 52.9% (49.1,55.9) | 261 (171,381) | 257 (144,373) | 326 (219,477) | 56.1% (41.0,76.7) |
| East Africa | 12.0 | 24.4% | 41.0% (38.4,43.5) | 21.1% (18.8,23.7) | 799 (661,1008) | 514 (415,633) | 42.8 (34.5,52.7) | 75.8 (61.4,91.9) | 55.2% (52.4,58.1) | 43.5% (41.3,46.1) | 181 (115,263) | 327 (236,429) | 476 (362,645) | 59.9% (49.7,70.2) |
| Southern Africa | 3.63 | 23.4% | 48.1% (46.2,50.4) | 33.3% (30.1,37.4) | 251 (206,317) | 205 (166,258) | 58.9 (47.5,74.1) | 124 (95.9,156) | 60.2% (56.4,63.5) | 48.4% (45.8,51.4) | 107 (70.9,165) | 96.3 (50.8,149) | 151 (105,208) | 60.7% (44.8,80.0) |
| West Africa | 16.3 | 23.0% | 56.6% (53.2,59.6) | 40.5% (35.7,44.3) | 1367(1118,1697) | 1128 (929,1410) | 69.2 (57.0,86.6) | 151 (127,190) | 60.8% (57.2,64.0) | 50.7% (47.2,53.7) | 535 (351,813) | 588 (343,835) | 773 (536,1083) | 57.2% (42.3,75.3) |
| **HBHI vs non-HBHI countries** | | | | | | | | | | | | | | |
| Non-HBHI | 18.1 | 24.1% | 38.1% (36.4,40.6) | 25.1% (22.6,26.9) | 1142 (942,1442) | 860 (700,1078) | 47.4 (38.6,59.4) | 91.9 (76.6,113) | 54.7% (52.3,57.1) | 46.5% (44.8,48.5) | 310 (200,454) | 540 (398,739) | 582 (423,788) | 51.9% (42.6,64.0) |
| HBHI | 23.7 | 21.6% | 60.0% (57.7,62.3) | 41.3% (38.7,44.5) | 1909(1564,2387) | 1558(1280,1924) | 65.8 (54.1,81.3) | 154 (126,188) | 61.3% (57.7,64.7) | 50.6% (47.3,53.1) | 785 (516,1158) | 779 (421,1129) | 1170 (812,1638) | 60.1% (45.6,79.1) |

[¥]Based on the impact of malaria upon Hb evaluated at 200 days gestation [±]Based upon impact of IPTp upon Hb evaluated during third trimester Abbreviations: **HBHI** – High burden High Impact, **WHO-AFRO** – World Health Organization Africa Region **Central Africa:** Angola, Cameroon, Central African Republic, Chad, DRC, Equatorial Guinea, Gabon, Republic of the Congo. **East Africa:** Burundi, Djibouti, Eritrea, Ethiopia, Kenya, Madagascar, Rwanda, Somalia, South Sudan, Tanzania, Uganda. **Southern Africa:** Botswana, Malawi, Mozambique, Namibia, South Africa, Zambia, Zimbabwe. **West Africa:** Benin, Burkina Faso, Côte d'Ivoire, The Gambia, Ghana, Guinea, Guinea-Bissau, Liberia, Mali, Niger, Nigeria, Senegal, Sierra Leone, Togo **African HBHI countries**: Burkina Faso, Cameroon, DRC, Ghana, Mali, Mozambique, Niger, Nigeria, Tanzania, Uganda **Non-HBHI African countries with sustained malaria transmission:** Angola, Botswana, Burundi, Central African Republic, Chad, Côte d'Ivoire, Djibouti, Eritrea, Ethiopia, Gabon, The Gambia, Guinea, Guinea-Bissau, Kenya, Liberia, Madagascar, Malawi, Namibia, Rwanda, Senegal, Sierra Leone, Somalia, South Africa, South Sudan, Sudan, Togo, Zambia, Zimbabwe.

# Reporting Summary

## Statistics

For all statistical analyses, confirm that the following items are present in the figure legend, table legend, main text, or Methods section.

| n/a | Confirmed | |
|---|---|---|
| ☐ | ☒ | The exact sample size (*n*) for each experimental group/condition, given as a discrete number and unit of measurement |
| ☐ | ☒ | A statement on whether measurements were taken from distinct samples or whether the same sample was measured repeatedly |
| ☐ | ☒ | The statistical test(s) used AND whether they are one- or two-sided<br>*Only common tests should be described solely by name; describe more complex techniques in the Methods section.* |
| ☐ | ☒ | A description of all covariates tested |
| ☐ | ☒ | A description of any assumptions or corrections, such as tests of normality and adjustment for multiple comparisons |
| ☐ | ☒ | A full description of the statistical parameters including central tendency (e.g. means) or other basic estimates (e.g. regression coefficient) AND variation (e.g. standard deviation) or associated estimates of uncertainty (e.g. confidence intervals) |
| ☒ | ☐ | For null hypothesis testing, the test statistic (e.g. *F*, *t*, *r*) with confidence intervals, effect sizes, degrees of freedom and *P* value noted<br>*Give P values as exact values whenever suitable.* |
| ☐ | ☒ | For Bayesian analysis, information on the choice of priors and Markov chain Monte Carlo settings |
| ☐ | ☒ | For hierarchical and complex designs, identification of the appropriate level for tests and full reporting of outcomes |
| ☒ | ☐ | Estimates of effect sizes (e.g. Cohen's *d*, Pearson's *r*), indicating how they were calculated |

*Our web collection on statistics for biologists contains articles on many of the points above.*

## Software and code

Policy information about availability of computer code

| Data collection | Our analysis involved no primary data collection. |
|---|---|
| Data analysis | All code used for this analysis is open access and available via the following repositories:<br><br>Hemoglobin dynamics and anemia risk model: https://github.com/patrickgtwalker/Anaemia_malaria_fitting_open<br><br>Integrated malaria in pregnancy transmission model: https://github.com/patrickgtwalker/malaria_in_pregnancy_istp_model_open<br><br>Both repositories include full documentation and implementation guidance. These resources provide all code necessary to reproduce the main analyses and figures. Feedback and contributions are welcome. |

For manuscripts utilizing custom algorithms or software that are central to the research but not yet described in published literature, software must be made available to editors and reviewers. We strongly encourage code deposition in a community repository (e.g. GitHub). See the Nature Portfolio guidelines for submitting code & software for further information.

## Data

Policy information about availability of data

All manuscripts must include a data availability statement. This statement should provide the following information, where applicable:
- Accession codes, unique identifiers, or web links for publicly available datasets
- A description of any restrictions on data availability
- For clinical datasets or third party data, please ensure that the statement adheres to our policy

The dataset from Tagbor et al. (2015) is archived at the LSHTM Data Compass repository (DOI: 10.17037/DATA.2) and is available under controlled access upon request.

The dataset from Madanitsa et al. (2016) is available via the WorldWide Antimalarial Resistance Network (WWARN), with access subject to review by the WWARN Data Access Committee.

Individual participant-level data from Desai et al. (2015) and Madanitsa et al. (2023) are available from the study investigators upon reasonable request and subject to appropriate data use agreements and ethical approvals.

All processed data outputs used to generate model results (e.g., hemoglobin trajectories, anemia risk estimates, IPTp impact) are included in the Supplementary Information. All secondary data sources—including malaria prevalence surfaces (Malaria Atlas Project), demographic and fertility data (DHS/MIS), and population estimates (WorldPop)—are publicly available and cited in the manuscript.

## Research involving human participants, their data, or biological material

Policy information about studies with human participants or human data. See also policy information about sex, gender (identity/presentation), and sexual orientation and race, ethnicity and racism.

| | |
|---|---|
| Reporting on sex and gender | All trials enrolled pregnant individuals and included data on gravidity (parity), but not on gender identity or presentation. All participants were biologically female by study design. |
| Reporting on race, ethnicity, or other socially relevant groupings | Race and ethnicity were not recorded in any of the trials and were not used in the design, analysis, or interpretation of the study. Country of residence and site-specific malaria risk were used to capture geographic and epidemiologic heterogeneity. |
| Population characteristics | The study population comprises pregnant women attending antenatal care across sites in Burkina Faso, Ghana, The Gambia, Mali, Kenya, Malawi, and Tanzania between 2010 and 2019. Participants were enrolled during the second trimester (14–27 weeks' gestation) as part of four randomized controlled trials of malaria prevention in pregnancy and followed through delivery. |
| Recruitment | Participants were recruited through antenatal clinics in the context of ethically approved clinical trials. All participants provided written informed consent prior to enrollment. Trial-specific eligibility criteria, randomization procedures, and clinical endpoints are detailed in the original publications. |
| Ethics oversight | This study is a secondary analysis of de-identified individual-level data from four previously published randomized trials. Each trial was approved by the relevant national research ethics committees and institutional review boards in participating countries. Written informed consent was obtained from all participants at the time of enrollment. This secondary analysis was conducted with the permission of the original trial investigators and adheres to all applicable ethical and data protection standards. No new participants were enrolled, and no additional data collection was undertaken. |

Note that full information on the approval of the study protocol must also be provided in the manuscript.

# Field-specific reporting

Please select the one below that is the best fit for your research. If you are not sure, read the appropriate sections before making your selection.

☒ Life sciences ☐ Behavioural & social sciences ☐ Ecological, evolutionary & environmental sciences

For a reference copy of the document with all sections, see nature.com/documents/nr-reporting-summary-flat.pdf

# Life sciences study design

All studies must disclose on these points even when the disclosure is negative.

| | |
|---|---|
| Sample size | No new participants were enrolled for this secondary analysis. The sample size (n = 12,608) reflects the total number of pregnant women enrolled in four previously conducted randomized controlled trials in sub-Saharan Africa. Sample size in each trial was predetermined based on the original clinical objectives and statistical power calculations as described in the trial protocols and primary publications. All available participant data from these trials were included in the analysis. |
| Data exclusions | No additional exclusion criteria were applied beyond those specified in the original trial protocols. In some trials, exclusions were applied for biological or protocol-driven reasons—such as restriction to specific gravidity groups or exclusion of women with hemoglobin <7 g/dL at |

enrollment due to concerns about trial safety. Our analytic approach explicitly adjusted for these exclusions to ensure that the resulting estimates are representative of the broader population of antenatal care attendees. Full details of the trial-specific exclusion criteria and how they were accounted for in the analysis are provided in the Methods and Supplementary Information.

| | |
|---|---|
| Replication | This study used previously published datasets and did not include experimental replication. Reproducibility of the findings was evaluated by testing model fit across multiple trials and sites, as well as by reproducing observed hemoglobin trajectories and anemia prevalence patterns. All code used for model fitting and replication is available in public GitHub repositories. |
| Randomization | All participants included in the analysis were originally enrolled in randomized controlled trials. Randomization procedures—including group allocation methods and control of potential confounders—were determined by each trial's design and are detailed in their original publications. |
| Blinding | Blinding of participants and study staff was implemented in the original trials according to their respective protocols (e.g., double-blind, placebo-controlled designs). |

# Reporting for specific materials, systems and methods

We require information from authors about some types of materials, experimental systems and methods used in many studies. Here, indicate whether each material, system or method listed is relevant to your study. If you are not sure if a list item applies to your research, read the appropriate section before selecting a response.

## Materials & experimental systems

| n/a | Involved in the study |
|---|---|
| ☒ | ☐ Antibodies |
| ☒ | ☐ Eukaryotic cell lines |
| ☒ | ☐ Palaeontology and archaeology |
| ☒ | ☐ Animals and other organisms |
| ☐ | ☒ Clinical data |
| ☒ | ☐ Dual use research of concern |
| ☒ | ☐ Plants |

## Methods

| n/a | Involved in the study |
|---|---|
| ☒ | ☐ ChIP-seq |
| ☒ | ☐ Flow cytometry |
| ☒ | ☐ MRI-based neuroimaging |

## Clinical data

Policy information about clinical studies

All manuscripts should comply with the ICMJE guidelines for publication of clinical research and a completed CONSORT checklist must be included with all submissions.

| | |
|---|---|
| Clinical trial registration | This study is a secondary analysis of de-identified individual-level data from four randomized controlled trials of malaria in pregnancy, all prospectively registered:<br>Tagbor et al. (2015) — NCT01084213<br>Desai et al. (2015) — NCT01669941<br>Madanitsa et al. (2016) — Pan AfricanClinical Trials Registry PACTR201103000280319;ISRCTNRegistry ISRCTN69800930<br>Madanitsa et al., (2023) — NCT03208179<br>Each trial was conducted under ethical oversight by recognized national and institutional review boards. |
| Study protocol | This study involved secondary analysis of anonymized individual participant data from four randomized controlled trials of malaria prevention in pregnancy. All trials enrolled pregnant women through antenatal care (ANC) clinics in sub-Saharan Africa and recorded hemoglobin concentration, malaria status, gestational age, and gravidity at enrollment. No new data collection or participant enrolment was conducted for this analysis. Trial-specific inclusion and exclusion criteria were accounted for to ensure population representativeness.<br><br>Madanitsa et al. (2023): A double-blind, randomized, partly placebo-controlled trial conducted in Kenya, Malawi, Tanzania, and Burkina Faso. Women aged ≥15 years at 16–28 weeks' gestation were randomized to monthly sulfadoxine–pyrimethamine (SP), dihydroartemisinin–piperaquine (DP), DP with azithromycin (AZ), or placebo (in selected sites). Key exclusion criteria included multiple pregnancy, chronic illness, and contraindications to study drugs. Women were not excluded based on hemoglobin concentration.<br><br>Madanitsa et al. (2016): A randomized trial in southern Malawi comparing monthly IPTp with SP versus DP. Participants were enrolled via ANC and followed through delivery. Women with Hb <7 g/dL were excluded at baseline.<br><br>Desai et al. (2015): An open-label, three-arm randomized superiority trial conducted in western Kenya. Women at 16–32 weeks' gestation were randomized to IPTp-SP, ISTp-DP, or IPTp-DP. Primary outcomes included maternal malaria and anemia. Women with Hb <7 g/dL were excluded.<br><br>Tagbor et al. (2015): A three-arm randomized controlled non-inferiority trial conducted in Burkina Faso, Mali, The Gambia, and Ghana. Women were randomized to IPTp-SP, ISTp-SP, or ISTp-AS–AQ and followed monthly through delivery. The trial enrolled only primigravidae and secundigravidae. No exclusion based on hemoglobin concentration was applied. |

All original trials received ethics approval from local and collaborating institutional review boards, and written informed consent was obtained from all participants at the time of enrolment.

Data collection

Individual-level data were obtained from four randomized controlled trials conducted between 2010 and 2019 in seven sub-Saharan African countries. All data used in this analysis were collected at the point of enrolment during antenatal care (ANC) visits, prior to administration of intermittent preventive treatment. Key variables included hemoglobin concentration, malaria infection status (by PCR or microscopy), gestational age, and gravidity.

Madanitsa et al., 2016 (Malawi IST): Enrolment took place between 2011 and 2013 in southern Malawi. Women were enrolled at 14–26 weeks' gestation.

Desai et al., 2015 (Kenya IST): Conducted in western Kenya from 2012 to 2014. Women were enrolled at 16–32 weeks' gestation.

Tagbor et al., 20105(West Africa trial): Participants were enrolled between 2010 and 2011 in Ghana, Burkina Faso, Mali, and The Gambia. Only primigravidae and secundigravidae were eligible.No gestational limts on enrollment were applied.

Madanitsa et al., 2023 (IMPROVE-1): Enrolment occurred between 2018 and 2019 in Kenya, Malawi, and Tanzania. Women were enrolled at 16–28 weeks' gestation.

Outcomes

The primary outcome for our secondary analysis was the prevalence of malaria-attributable maternal anemia, stratified by severity (moderate: Hb <9 g/dL; severe: Hb <7 g/dL) and gestational age. These thresholds were pre-specified based on established clinical risk levels for postpartum hemorrhage and maternal mortality. Hemoglobin concentration at enrolment was recorded using standard clinical protocols in all four source trials. Malaria status was determined via PCR (or microscopy where PCR was unavailable), and gestational age was based on last menstrual period or fundal height as recorded at enrolment.

# Plants

Seed stocks

*Report on the source of all seed stocks or other plant material used. If applicable, state the seed stock centre and catalogue number. If plant specimens were collected from the field, describe the collection location, date and sampling procedures.*

Novel plant genotypes

*Describe the methods by which all novel plant genotypes were produced. This includes those generated by transgenic approaches, gene editing, chemical/radiation-based mutagenesis and hybridization. For transgenic lines, describe the transformation method, the number of independent lines analyzed and the generation upon which experiments were performed. For gene-edited lines, describe the editor used, the endogenous sequence targeted for editing, the targeting guide RNA sequence (if applicable) and how the editor was applied.*

Authentication

*Describe any authentication procedures for each seed stock used or novel genotype generated. Describe any experiments used to assess the effect of a mutation and, where applicable, how potential secondary effects (e.g. second site T-DNA insertions, mosiacism, off-target gene editing) were examined.*

