## [Peer Review File · Nature Health]

The burden of malaria-attributable maternal anemia and the impact of preventive treatment across sub-Saharan Africa

Corresponding Author: Dr Patrick Walker

Version 0:

Reviewer comments:

Reviewer #1

(Remarks to the Author)

This study addresses an important public health issue by providing updated estimates and quantitative insights on the burden caused by malaria in pregnancy, which can lead to severe anemia and other complications. The study focuses on sub-Saharan Africa, where approximately 40% of maternal deaths are linked to hemorrhage and severe anemia, and also evaluates the impact of intermittent preventive treatment with sulfadoxine–pyrimethamine in these countries. More specifically, the authors used a mathematical model of malaria exposure during pregnancy to quantify the risk of anemia, accounting for immunity acquired through successive infections, varying levels of severity based on hemoglobin concentration thresholds, and incorporating spatiotemporal data on malaria prevalence, population density, and fertility patterns. The link between malaria exposure and hemoglobin concentration is modeled using a sophisticated statistical framework. Using this approach, the authors estimated the burden of malaria-attributable anemia in the absence of pregnancy-specific interventions and assessed the extent to which this risk is currently mitigated by IPTp. While I am not an expert in hierarchical Bayesian models, the manuscript presents the key modeling ideas and assumptions clearly. Overall, the approach appears technically sound and robust, and the limitations are well described. The manuscript is well-written, and the findings are highly relevant for policy and public health planning. I therefore have only a few minor points to suggest.

Although immunity acquired from infections outside of pregnancy may differ from that developed during pregnancy, I would be curious to see some exploration of this, as the data might allow it. For example, I would expect that an increase in maternal age (in years) could indicate higher accumulated exposure to malaria outside of pregnancy, and I wonder how age influences the reduction in hemoglobin specifically for primigravidae women. Similarly, as a way to validate the method for inferring previous immunity based on the mathematical model, it might be interesting to check whether areas with higher malaria prevalence correspond to higher inferred immunity levels in primigravidae. While this is not crucial for the main aims of the paper, performing such checks could provide some confidence in how well the model reconstructs immunity

Additional minor points:

- Line 70: “Malaria infection during pregnancy also contributes” -> remove “also.”
- Line 159: A parenthesis and a period are missing after “(Figure S1”
- Lines 159–163: The authors refer to Figure 1a as representing “primigravid women.” However, in the caption, “other women experiencing their first infected pregnancy” are also included. Please clarify and unify the description.
- Figure 2: I suggest reordering the panels to match the narrative of the text, where Fig 2e is described before Fig 2b–c. A possible order could be a, e, b, c (upper panels) and d, h, f, g (bottom panels). Panels f and g may not be essential, as they are not discussed in the text. The x-axis of panels d and h is unclear: please describe them in the caption. Colors in panels d and h represent different variables compared to the other panels, so I suggest using a different color palette to improve clarity
- Line 219: “across Africa” should be changed to “sub-Saharan Africa” or “malaria-endemic regions of Africa.”
- Figure 3: Could you add, at the bottom, a map showing the estimated current burden of malaria-attributable severe anemia (i.e., number of cases) under the interventions currently in place? This would provide visual support for the text reported at lines 241–243. This map could also be complemented by a map showing the estimated overall per-pregnancy risk of severe anemia, thereby supporting the text reported at lines 246–253.
- Lines 255–257: “To quantify the extent to which reductions in malaria transmission in the 21st century have mitigated the risk of maternal anemia through decreased exposure during pregnancy, we estimated the number of malaria-attributable anemia cases.” I think here you are referring to the number of malaria-attributable anemia cases expected in the absence of interventions. Please clarify.
- Figure 4: The text “PfPr2–10%” seems to be misplaced. I suggest repeating it for both figures composing panel b.
- Lines 313: “We then assumed that IPTp-SP mitigates this reduction by a constant proportion across settings.” This seems a strong assumption given the figures showing strong heterogeneity across countries and may represent a potential limitation of the study.

- Lines 314–316: Is the 53.7% reduction estimated by the model applied to both primigravidae and multigravidae women? Given results shown in Figure 5a, I do not think so. Please clarify.
- Caption Figure 5: Please define HBHI at its first occurrence for readers not familiar with this wording. Panel 5d: The figure shows estimated cases, not averted ones. Also, there is a misalignment between the caption and the figure label. Does panel 5d show only severe anemia or also moderate anemia? Based on the numbers in the text, I would say severe only.
- Line 345: A space is missing in the text (“see Methods),we”).
- Line 538 [Equation 1]: Symbols in the equation are missing/misprinted, and the equation is not visible.

(Remarks on code availability)

Reviewer #2

(Remarks to the Author)

Review Leuba et al 2025 Nature Health. The burden of malaria-attributable maternal anemia and the impact of IPTp across 1 sub-Saharan Africa

This manuscript gives a spotlight to the under-reported importance of the effect of maternal malaria on anaemia and improves understanding of the interaction between malaria and maternal anaemia and malaria prevention. The authors take us stepwise through different scenarios with models based on enrolment data from four trials in nine study sites. I would like to complement the authors on the undertaking of this effort and bringing this together in a manuscript that shows how important it is and will be to continue malaria prevention.

Abstract

The abstract could be improved by adding the definition of moderate or severe anaemia.

The last sentence could perhaps be replaced: IPTp with SP is known not to be effective for malaria prevention in all African regions, so perhaps they could rephrase as “Sustaining effective malaria prevention...” which may not necessarily be IPTp but could also include other ways to reduce malaria.

Introduction

The introduction offers a very nice summary of the problem of anaemia in pregnancy. The relationship between anaemia and haemorrhage has been severely underreported, and it is good this gets attention.

This sentence is not clear, perhaps split?

“We first estimated the “intrinsic risk”, defined here as the burden of malaria-attributable anemia that would occur in the absence of pregnancy-specific interventions such as IPTp-SP, which has changed in response to declining population-level transmission following malaria control scale-up since the early 2000s.”

Because results in this journal come before the methods section, it may increase clarity to introduce used definitions of anaemia in the introduction, instead of halfway the results.

Results

Please note that there are several definitions used for the upper limit of anaemia, e.g. <10.5 or 11 g/dl can be found.

However, 10 g/dl as the upper border is not common, and the reference used here is old (Kavle 2008). I think the references are off; however, even the WHO guideline referred to in ref 33 use 10.5 and 11 g/dl as upper limit. So please help the reader and introduce the definitions used in abstract and introduction (because methods come after the results).

The abbreviation Cri is not introduced in the main text (or I missed it).

Figure S1 is very interesting but a bit hard to read and assess and compare. Perhaps the author can add reference lines for normal value of haemoglobin (e.g. 10.5 g/dl) as a light coloured or dotted line, so you can assess how the trend is across transmission levels. E.g. is it now hard to read if the haemoglobin among G4 is decreasing or increasing across transmission levels. I just want to check if the mean haemoglobin is actually lower in low malaria prevalence countries, which would be kind of counter intuitive.

“This model indicated that in uninfected women, early pregnancy Hb levels increase with gravidity and decline steadily across all gravidity groups during the second trimester (Figure S1).” I don’t see any upward going lines, only downward in S1?

I am also confused by Figure 1c, where is talked about malaria associated reduction in Hb by transmission region. If the y-point is 0, this means there is no reduction in haemoglobin with malaria? And if the y-point is -1 does that mean there is an increase in haemoglobin with malaria (e.g. for G5, and G6+), which is counterintuitive? Or are these just small samples?

Line 361 : ref missing?

“...a classification used to prioritize malaria control efforts in the ten African countries with the highest absolute malaria burden [ref].”

Methods

I am confused about the use of PCR. PCR can be a good indicator of transmission, but the burden of anaemia has been

generally assessed using microscopy and not PCR. The association between PCR and anaemia is likely to be less strong compared to microscopy and RDT (because of lower parasite densities). I am not sure how the association between IPTp-SP and anaemia, as measured among three studies in Kenya among mostly women with low gravidity number, is representative for East and West Africa. MAP also provides data derived from malaria by microscopy to my knowledge.

There is now evidence that IPTp may also have beneficial effects unrelated to malaria. How could your data refute or show that?

Supplement:

There seems to be plenty of space below Figure S2 to explain what S, P T, D A and U are (and r)

Typos:

Line 159 "(Figure S1In"

For clarity and to avoid confusion for "G1 prevalence at 1st ANC", perhaps you can add "G1 malaria (PCR) at 1st ANC" to clarify that this is meant to be an indicator of malaria transmission level, and nothing to do with anaemia.

(Remarks on code availability)

Version 1:

Reviewer comments:

Reviewer #1

(Remarks to the Author)

The authors have thoroughly addressed most of my previous comments and substantially strengthened the manuscript.

Although the assumption of a constant proportional effect of IPTp-SP has been clarified, the authors should more clearly acknowledge as a key limitation the fact that their model does not account for the substantial heterogeneity in SP resistance and IPTp efficacy across regions. This variability, evident in both the figures and the trial data, is not explicitly explored or quantified, and its omission should be more clearly discussed.

In addition, one point remains only indirectly addressed: my suggestion to explore whether areas with higher malaria prevalence correspond to higher inferred immunity levels in primigravidae. While the additional age-gravidity analysis is valuable, it does not directly test this spatial consistency, which could still provide a useful validation of the model's inferred immunity component, even if only descriptively.

(Remarks on code availability)

Reviewer #2

(Remarks to the Author)

The authors have responded to my queries in a satisfying manner, and I think the manuscript is acceptable for publication.

(Remarks on code availability)

We thank both reviewers for their thoughtful, constructive, and encouraging evaluations. Their comments strengthened the manuscript scientifically and improved clarity throughout. In response, we have revised the text and figures accordingly, with the most substantive changes as follows:

- Age vs. gravidity disentanglement: Added new multivariable mixed-effects analyses jointly modelling age and gravidity (gestational age via natural cubic splines; site random intercepts). Age improves baseline Hb fit, but the malaria×age interaction (conditional on gravidity) is minimal; gravidity-specific malaria effects remain robust. Results and statistics are now reported and referenced (Supplementary Fig. S3), directly addressing the reviewer's request. We believe that this really strengthens the manuscript and provides new insight within the wider literature of drivers of malaria in pregnancy burden so are particularly grateful for this suggestion.
- Definition and early placement of Hb thresholds: Introduced anemia thresholds (including the <9 g/dL definition) in the Abstract/early Results, with a clearer rationale linking this cut-off to clinically meaningful risk and to sensitivity in the lower tail of the Hb distribution.
- Figure clarifications and guidance: Clarified Figure 1a wording (who is pregnancy-naïve and why primigravidae anchor the comparison) and streamlined the caption; added threshold guide lines (7, 9, 10 g/dL) and clearer narration for Figure 2 ordering and colour schemes; corrected panel 5d labeling and defined HBHI at first occurrence; improved Figure 4 labelling (spelled-out PfPR2–10).
- Counterfactual language: Made explicit that the 2000 vs 2023 comparison quantifies intrinsic burden under a no-intervention counterfactual using MAP surfaces, and signposted Figure 4 at first mention of spatial burden.
- IPTp efficacy assumption: Clarified that IPTp-SP mitigates a constant proportion of malaria-attributable Hb loss across settings; thus absolute benefits are largest in paucigravidae and negligible in multigravidae with prior exposure, aligning well with all empirical data available from the most recent Cochrane review. We also amended text in the Discussion noting potential heterogeneity (e.g., SP resistance, non-malarial effects) as a limitation.
- PCR vs microscopy rationale: Expanded justification for using PCR-detected infection to capture sub-patent infections relevant to pregnancy-specific immunity and to avoid bias in multigravidae; noted incorporation of MAP microscopy surfaces within our burden framework.
- Edits for precision and readability: Corrected minor textual issues (typos, punctuation), harmonised terminology (e.g., “malaria-endemic regions of Africa”), fixed equation rendering.

In the following we provide a point-by-point response to each comment. We again thank the reviewers for their efforts in helping us to substantially improve our manuscript.

Reviewer #1 (Remarks to the Author):

This study addresses an important public health issue by providing updated estimates and quantitative insights on the burden caused by malaria in pregnancy, which can lead to severe anemia and other complications. The study focuses on sub-Saharan Africa, where approximately 40% of maternal deaths are linked to hemorrhage and severe anemia, and also evaluates the impact of intermittent preventive treatment with sulfadoxine–pyrimethamine in these countries. More specifically, the authors used a mathematical model of malaria exposure during pregnancy to quantify the risk of anemia, accounting for immunity acquired through successive infections, varying levels of severity based on hemoglobin concentration thresholds, and incorporating spatiotemporal data on malaria prevalence, population density, and fertility patterns. The link between malaria exposure and hemoglobin concentration is modeled using a sophisticated statistical framework. Using this approach, the authors estimated the burden of malaria-attributable anemia in the absence of pregnancy-specific interventions and assessed the extent to which this risk is currently mitigated by IPTp. While I am not an expert in hierarchical Bayesian models, the manuscript presents the key modeling ideas and assumptions clearly. Overall, the approach appears technically sound and robust, and the limitations are well described. The manuscript is well-written, and the findings are highly relevant for policy and public health planning. I therefore have only a few minor points to suggest.

We thank the reviewer for a careful and constructive evaluation of our work.

Although immunity acquired from infections outside of pregnancy may differ from that developed during pregnancy, I would be curious to see some exploration of this, as the data might allow it. For example, I would expect that an increase in maternal age (in years) could indicate higher accumulated exposure to malaria outside of pregnancy, and I wonder how age influences the reduction in hemoglobin specifically for primigravidae women. Similarly, as a way to validate the method for inferring previous immunity based on the mathematical model, it might be interesting to check whether areas with higher malaria prevalence correspond to higher inferred immunity levels in primigravidae. While this is not crucial for the main aims of the paper, performing such checks could provide some confidence in how well the model reconstructs immunity

We thank the reviewer for this insightful suggestion. Motivated by the comment, we conducted additional analyses to probe whether immunity accrued outside pregnancy (proxied by maternal age) independently modifies malaria-associated

hemoglobin loss, and whether doing so alters our gravidity findings. Specifically, we fit multivariable mixed-effects models including both maternal age and gravidity (gestational age modeled with a natural cubic spline; random intercept for study site). Adding age improved overall fit—consistent with higher baseline hemoglobin at older ages—yet the interaction between malaria and age, conditional on gravidity, was effectively null (likelihood-ratio test $\chi^2=31.8$ for adding age main effect vs. $\chi^2=7.6$, $p=0.48$ for adding the three-way interaction), indicating that hemoglobin increases with age at a similar rate in infected and uninfected women. Importantly, the gravidity pattern remained robust when age was included: gravidity-specific malaria effects were stable and changed minimally under age adjustment. These results are presented in Supplementary Figure 3 and we believe provide a substantial addition to both the analysis and a contribution to body of research showing the primary role of pregnancy-specific immunity in shaping the impact of malaria upon pregnancy outcome. In addition, we now situate our inference within more recent external immuno-epidemiological evidence showing that pregnancy-specific immunity—protective against placental sequestration—accumulates over successive infected pregnancies and that placental malaria-associated antibody acquisition exhibits a stronger gradient by gravidity at higher transmission intensity (Matambisso, BMC Medicine 2022) which is consistent with our interpretation

Additional minor points:

- Line 70: “Malaria infection during pregnancy also contributes” -> remove “also.”

Corrected

- Line 159: A parenthesis and a period are missing after “(Figure S1”

Corrected

- Lines 159–163: The authors refer to Figure 1a as representing “primigravid women.” However, in the caption, “other women experiencing their first infected pregnancy” are also included. Please clarify and unify the description.

We appreciate the reviewer’s careful reading and for highlighting a seeming ambiguity. In our model, the malaria-attributable reduction in haemoglobin (g/dL) is the same for all women with no prior exposure in pregnancy. We compare these estimates with primigravidae in Figure 1a, as they are the only group in these data (and in most other datasets) who can be unambiguously identified as pregnancy-naïve. By contrast, secundigravidae represent a mixture of women who are still immunologically naïve and those already exposed, which is why both our model fit and the observed data show a declining average impact with increasing G1 malaria prevalence (Figure 1c). To avoid any possible confusion, we have clarified this point in the main text and streamlined the caption to read:

Posterior estimates (yellow lines) of the reduction in hemoglobin (Hb) resulting from malaria infection among women experiencing their first malaria-exposed pregnancy (including primigravidae and

multigravidae without prior exposure). Dots and error bars show observed differences in Hb between infected and uninfected primigravidae (the only group in these data for whom absence of prior malaria exposure in pregnancy can be unambiguously established) grouped by study-specific quintiles of gestational age at enrolment and plotted at the mean of each quintile

- Figure 2: I suggest reordering the panels to match the narrative of the text, where Fig 2e is described before Fig 2b–c. A possible order could be a, e, b, c (upper panels) and d, h, f, g (bottom panels). Panels f and g may not be essential, as they are not discussed in the text. The x-axis of panels d and h is unclear: please describe them in the caption. Colors in panels d and h represent different variables compared to the other panels, so I suggest using a different color palette to improve clarity

We thank the reviewer for these helpful suggestions. We considered reordering the panels but decided to retain the current layout, as it provides a consistent and visually clear structure: the top row presents risk per 1,000 infected pregnancies, while the bottom row presents risk per 1,000 total pregnancies. Within each row, panels are ordered by gravidity (G1, G2, G3+, all women), which we believe aids comparison across groups. To pre-empt confusion, we have revised the main text to explicitly guide the reader through the figure in this order, and the caption now states:

“Figure 2 shows risk per 1,000 infected pregnancies (top row, panels a–d ordered by gravidity category: G1, G2, G3+, all women) and risk per 1,000 total pregnancies (bottom row, panels e–h in the same order), to aid comparison.”

We have ensured d and h have a different colour scheme to the rest in line with the reviewer’s suggestion.

- Line 219: “across Africa” should be changed to “sub-Saharan Africa” or “malaria-endemic regions of Africa.”

Amended to “malaria-endemic regions of Africa.”

- Figure 3: Could you add, at the bottom, a map showing the estimated current burden of malaria-attributable severe anemia (i.e., number of cases) under the interventions currently in place? This would provide visual support for the text reported at lines 241–243. This map could also be complemented by a map showing the estimated overall per-pregnancy risk of severe anemia, thereby supporting the text reported at lines 246–253.

We thank the reviewer for this suggestion. The information requested is already presented in Figure 4, which shows the spatial distribution of severe anaemia intrinsic risk and are grateful for the opportunity to signpost this better within the manuscript. Absolute burden maps (i.e. number of cases) are less informative for understanding spatial heterogeneity, as they are dominated by population density, but we do provide population-weighted estimates of burden by transmission intensity

(both for 2023 and 2000) in Figure 4b. To avoid any ambiguity, we have now signposted Figure 4 more explicitly in the Results text at the relevant point:

“By integrating our Hb-based risk model with a spatially explicit transmission model, we estimated the continent-wide burden of malaria-attributable maternal anemia across malaria-endemic regions of Africa, accounting for local variation in transmission intensity and fertility patterns (see Figure 4).”

• Lines 255–257: “To quantify the extent to which reductions in malaria transmission in the 21st century have mitigated the risk of maternal anemia through decreased exposure during pregnancy, we estimated the number of malaria-attributable anemia cases.” I think here you are referring to the number of malaria-attributable anemia cases expected in the absence of interventions. Please clarify.

We thank the reviewer for pointing out this ambiguity. We have revised the text to make it explicit that these estimates refer to a counterfactual, no-intervention scenario. The sentence now reads:

“To quantify the extent to which reductions in malaria transmission in the 21st century have mitigated the intrinsic risk of maternal anemia through decreased exposure during pregnancy, we estimated the number of malaria-attributable anemia cases that would have occurred in a counterfactual scenario in which pregnancies in 2023 experienced transmission levels observed in 2000.”

• Figure 4: The text “PfPr2–10%” seems to be misplaced. I suggest repeating it for both figures composing panel b.

This was supposed to span both figures as a shared label but on reflection did look a bit small/hanging. We have now better utilized the space by writing out the abbreviation in full and we believe this optimises overall clarity beyond repeating the abbreviated version twice.

• Lines 313: “We then assumed that IPTp-SP mitigates this reduction by a constant proportion across settings.” This seems a strong assumption given the figures showing strong heterogeneity across countries and may represent a potential limitation of the study.

We address both points re lines 313-316 after the following comment.

• Lines 314–316: Is the 53.7% reduction estimated by the model applied to both primigravidae and multigravidae women? Given results shown in Figure 5a, I do not think so. Please clarify.

We thank the reviewer for raising this important point and for the opportunity to clarify. Our assumption is that IPTp-SP mitigates the **malaria-attributable reduction in haemoglobin by a constant proportion**, not that it produces the same absolute effect across all groups. Because the absolute malaria-attributable impact is much larger in primigravidae and secundigravidae, the same proportional reduction translates into a greater absolute benefit in these groups, and essentially no

detectable benefit in multigravidae with prior exposure. This matches both the quantitative results of the trials included in our fitting and the qualitative findings of trials that excluded anaemic women at enrolment (which would bias direct comparisons but which consistently showed no effect in higher gravidities).

To avoid misunderstanding, we have revised the relevant passage to read:

“...we then assumed that IPTp-SP mitigates the malaria-attributable reduction in haemoglobin by a constant proportion across settings, such that the absolute benefit is greatest in primigravidae and declines with increasing gravidity as the malaria-attributable effect itself becomes smaller.”

We have also included the following paragraph in our discussion which notes we do not capture potential heterogeneities:

“Our analysis does not account for potential changes in IPTp effectiveness due to evolving SP resistance since the initial trials. The extent to which such resistance undermines the protective effect of IPTp-SP on anemia remains under scientific debate⁵¹, further complicated by SP’s potential effects on non-malarial causes of maternal and neonatal morbidity⁵², which our study does not attempt to quantify. However, no alternative intervention has demonstrated consistently superior performance in preventing malaria in pregnancy. Any reduction in efficacy due to resistance does not diminish the need for continued IPTp delivery—on the contrary, it heightens the risk posed by any future resurgence in malaria transmission, potentially compounding maternal and neonatal anemia-related burden.”

- Caption Figure 5: Please define HBHI at its first occurrence for readers not familiar with this wording.

Added

Panel 5d: The figure shows estimated cases, not averted ones. Also, there is a misalignment between the caption and the figure label. Does panel 5d show only severe anemia or also moderate anemia? Based on the numbers in the text, I would say severe only.

We are extremely grateful to the reviewer for spotting these errors. We have now corrected the caption and the reviewer is indeed correct that 5d refers to severe cases only.

- Line 345: A space is missing in the text (“see Methods),we”).

Corrected

- Line 538 [Equation 1]: Symbols in the equation are missing/misprinted, and the equation is not visible.

This was to do with the conversion from word doc to pdf. Hopefully this is now

corrected but we will review after conversion upon resubmission and if it still an issue will liaise with editorial team.

Reviewer #2 (Remarks to the Author):

Review Leuba et al 2025 Nature Health. The burden of malaria-attributable maternal anemia and the impact of IPTp across 1 sub-Saharan Africa

This manuscript gives a spotlight to the under-reported importance of the effect of maternal malaria on anaemia and improves understanding of the interaction between malaria and maternal anaemia and malaria prevention. The authors take us stepwise through different scenarios with models based on enrolment data from four trials in nine study sites. I would like to complement the authors on the undertaking of this effort and bringing this together in a manuscript that shows how important it is and will be to continue malaria prevention.

We thank the reviewer for their thoughtful and encouraging comments.

Abstract

The abstract could be improved by adding the definition of moderate or severe anaemia.

The last sentence could perhaps be replaced: IPTp with SP is known not to be effective for malaria prevention in all African regions, so perhaps they could rephrase as "Sustaining effective malaria prevention..." which may not necessarily be IPTp but could also include other ways to reduce malaria.

We have amended this sentence to:

"Sustaining protection for pregnant women from malaria is critical to prevent resurgence in severe maternal anemia as global malaria funding falters."

Introduction

The introduction offers a very nice summary of the problem of anaemia in pregnancy. The relationship between anaemia and haemorrhage has been severely underreported, and it is good this gets attention.

We agree that the contribution of anaemia to maternal haemorrhage has been under-recognised, and we are pleased that our framing helped to highlight this important link.

sentence is not clear, perhaps split?

"We first estimated the "intrinsic risk", defined here as the burden of malaria-attributable anemia that would occur in the absence of pregnancy-specific interventions such as IPTp-SP, which has changed in response to declining

population-level transmission following malaria control scale-up since the early 2000s.”

We thank the reviewer for this helpful suggestion. We have revised the sentence for clarity by splitting the definition of intrinsic risk from the description of how it has changed

“We first estimated the ‘intrinsic risk,’ defined as the burden of malaria-attributable anemia that would occur in the absence of pregnancy-specific interventions such as IPTp-SP. We then examined how this intrinsic risk has changed over time in response to declining population-level transmission following the scale-up of malaria control since the early 2000s.”

Because results in this journal come before the methods section, it may increase clarity to introduce used definitions of anaemia in the introduction, instead of halfway the results.

We have now made the thresholds used explicit in the abstract and the first section of the results section (i.e. as early as possible given we need to highlight some empirical results as part of our justification – see immediate comment below..) highlighting how the 9 g/dL threshold focused our analysis more effectively on those most at risk of incremental malaria-attributable burden.

Results

Please note that there are several definitions used for the upper limit of anaemia, e.g. <10.5 or 11 g/dl can be found. However, 10 g/dl as the upper border is not common, and the reference used here is old (Kavle 2008). I think the references are off; however, even the WHO guideline referred to in ref 33 use 10.5 and 11 g/dl as upper limit. So please help the reader and introduce the definitions used in abstract and introduction (because methods come after the results).

We thank the reviewer for raising the point about anaemia thresholds with reference to the <9 g/dL cut-off and agree this wasn’t clearly articulated within an early enough stage of the manuscript. We also wish to clarify why we adopted this definition in our analysis. Whilst broader thresholds for anaemia risk, such as Hb <10 g/dL, remain the standard for guiding clinical care, we noted that they did not map well to the incremental impact of malaria. In third-trimester primigravidae the median haemoglobin level across all studies (excluding those with anaemia-based exclusion criteria) was 10.7 g/dL. As a result, under a uniform 1 g/dL reduction, the “incremental cases” at the <10 g/dL cut-off are drawn mainly from women around the median (10–11 g/dL, ~32nd–62nd percentile) who shift into the ~25th percentile band (9–10 g/dL). These losses are important, but less clinically consequential than those occurring simultaneously among women already in the 9–10 g/dL range (~25th percentile), who are pushed below 9 g/dL into the ~10th percentile or lower. This

latter group is not captured by either a <7 g/dL or <10 g/dL threshold, but it is captured by a <9 g/dL definition. The <9 g/dL threshold also has clinical support, including a recent systematic review (Young et al. 2023) identifying Hb <9 g/dL as associated with substantially increased risk of postpartum haemorrhage and maternal complications.

Figure S1 is very interesting but a bit hard to read and assess and compare. Perhaps the author can add reference lines for normal value of haemoglobin (e.g. 10.5 g/dl) as a light coloured or dotted line, so you can assess how the trend is across transmission levels. E.g. is it now hard to read if the haemoglobin among G4 is decreasing or increasing across transmission levels. I just want to check if the mean haemoglobin is actually lower in low malaria prevalence countries, which would be kind of counter intuitive.

We have now added guidelines for thresholds of 7, 9 and 10 g/dL. Both empirical loess splines and model fits are stratified by uninfected and infected. For equivalent baseline uninfected curves we would expect infected Hb trajectories to be lower in lower transmission settings due to lower pregnancy specific immunity. Other effects influencing baseline (i.e. mean levels in non-infected women) are treated as a fixed effect, so whilst we have not identified any trend with transmission they would be reflected in our framework.

“This model indicated that in uninfected women, early pregnancy Hb levels increase with gravidity and decline steadily across all gravidity groups during the second trimester (Figure S1).” I don’t see any upward going lines, only downward in S1?

The upward trend in uninfected by gravidity is not obviously visible within Figure S1 which does not facilitate such comparison. Instead, the reader should have been pointed to Table S3 which gives the relevant coefficients (gamma parameters), on reflection these coefficient cease to be monotonic from around G3 onwards so we have now revised this sentence to:

“This model indicated that in uninfected women, early pregnancy Hb levels are higher within multigravidae than in primigravidae (Table S3) and decline steadily across all gravidity groups during the second trimester (Figure S1).”

I am also confused by Figure 1c, where is talked about malaria associated reduction in Hb by transmission region. If the y-point is 0, this means there is no reduction in haemoglobin with malaria? And if the y-point is -1 does that mean there is an increase in haemoglobin with malaria (e.g. for G5, and G6+), which is counterintuitive? Or are these just small samples?

The reviewer is correct on both counts. This figure demonstrates that in both the empirical data and the fitted model suggest the association between malaria and hemoglobin reduction is consistently seen across primigravidae in all trial sites. In multigravidae association decreases by transmission intensity and gravidity,

reflecting expected patterns of pregnancy-specific immunity acquisition. Negative point estimates are not significant (with Cis for one such datapoint missing from the original figure due to going beyond the y limits, now corrected) and, for those in blue, are likely somewhat affected by the selective exclusion of those who were anemic at the point of enrolment.

Line 361: ref missing?

“...a classification used to prioritize malaria control efforts in the ten African countries with the highest absolute malaria burden [ref].”

Corrected

Methods

I am confused about the use of PCR. PCR can be a good indicator of transmission, but the burden of anaemia has been generally assessed using microscopy and not PCR. The association between PCR and anaemia is likely to be less strong compared to microscopy and RDT (because of lower parasite densities).

We thank the reviewer for raising this important point regarding infection detection methods. We agree that microscopy has historically been the main metric used in assessing malaria-associated anaemia, reflecting the diagnostic tools available at the time. However, we believe that the use of the most sensitive methods available is optimal for an analysis such as this, which seeks to capture the relationship between malaria infection and anaemia, and this represents a key advance over traditional approaches. We have previously shown that pregnancy-specific immunity modulates the detectability of infection (Walker et al., Nat Commun 2020), which is highly correlated with gravidity and likely reflects the same immune processes we quantify here. Excluding sub-patent infections would risk biasing estimates of the relationship between infection and anaemia in multigravidae, as only higher-density infections would be captured—these are themselves likely markers of lower prior exposure. Including all detectable infections therefore allows us to more accurately disentangle the contribution of pregnancy-specific immunity and thus the likely impact of shifts in the broader transmission landscape in the past, present and future.

I am not sure how the association between IPTp-SP and anaemia, as measured among three studies in Kenya among mostly women with low gravidity number, is representative for East and West Africa.

We agree with the reviewer that additional data on the association between IPTp-SP and haemoglobin would be valuable. However, our results draw upon all available placebo-controlled trial data identified in the 2014 Cochrane review (Radeva-Petrova et al., 2014). As placebo-controlled comparisons can no longer be ethically conducted, these remain the best available evidence and were the basis for the original indication of IPTp-SP's impact on anaemia prevention. We would also note that this evidence base extends beyond Kenya and low-gravidity women. For example, the review includes studies in multigravidae (Farafenni, The Gambia;

Mbaye et al.) and in lower transmission settings outside Kenya (Uganda; Ndyomugenyi et al.). While we did not incorporate these directly into our formal fitting—due to potential bias from anaemia-based enrolment exclusions—they are presented in Figure 5a and are entirely consistent with our model estimates that there would have been very limited average impact on hemoglobin due to malaria in these trial populations.

MAP also provides data derived from malaria by microscopy to my knowledge. MAP representatives are co-authors to this analysis which incorporates spatial surfaces of microscopy within our burden estimates, using a conversion approach to ANC1 PCR prevalence and within our approach to MiP burden modelling which feeds into WHO's annual *World Malaria Report*.

There is now evidence that IPTp may also have beneficial effects unrelated to malaria. How could your data refute or show that?

We agree that several studies have pointed to potential non-malarial benefits of IPTp. However, our analysis is focused specifically on malaria-attributable anaemia. The gravidity-dependent effects observed in the Cochrane review, with no significant haemoglobin impact outside of paucigravidae, strongly suggest that malaria is the primary driver of IPTp-SP's effect on anaemia in these data. We acknowledge that additional benefits could arise through other mechanisms—for example, effects on birth outcomes or in populations with higher prevalence of non-malarial, IPTp-sensitive risk factors—but disentangling these pathways lies beyond the scope of the present work.

We have signposted potential role of IPTp-SP whilst highlighting that we do not attempt to address this issue within the analysis:

“The extent to which such resistance undermines the protective effect of IPTp-SP on anemia remains under scientific debate⁵¹, further complicated by SP's potential effects on non-malarial causes of maternal and neonatal morbidity⁵², which our study does not attempt to quantify.”

Supplement:

There seems to be plenty of space below Figure S2 to explain what S, P T, D A and U are (and r)

We agree this Figure wasn't sufficiently captioned. We have now provided a fuller explanation of the model structure, including the variables highlighted.

Typos:

Line 159 “(Figure S1In”

Corrected

For clarity and to avoid confusion for “G1 prevalence at 1st ANC”, perhaps you can add “G1 malaria (PCR) at 1st ANC” to clarify that this is meant to be an indicator of malaria transmission level, and nothing to do with anaemia.

We concur and have amended the manuscript thusly.

Reviewer #1 (Remarks to the Author):

The authors have thoroughly addressed most of my previous comments and substantially strengthened the manuscript.

We thank the reviewer for this positive assessment. As noted previously, the improvements are largely due to their thoughtful and constructive feedback.

Although the assumption of a constant proportional effect of IPTp-SP has been clarified, the authors should more clearly acknowledge as a key limitation the fact that their model does not account for the substantial heterogeneity in SP resistance and IPTp efficacy across regions. This variability, evident in both the figures and the trial data, is not explicitly explored or quantified, and its omission should be more clearly discussed.

We agree that this limitation had not been fully articulated. Model fits will rarely capture all study-specific heterogeneity, and although we believe that the constant-effect assumption performs well in reproducing the pronounced impact of IPTp in primigravidae in high-transmission settings, and its limited impact in multigravidae, we recognise that judgments about absolute versus relative fit can remain partly subjective. This is especially true when, as here, some datapoints appear qualitatively consistent with the model's structure (for example, the Mbaye et al. trial in Gambian multigravidae, which reported a non-significant negative effect of IPTp-SP on Hb) but cannot be incorporated formally due to trial-specific design constraints (here the exclusion of anemic women at enrolment).

Nonetheless, we fully agree that numerous additional factors may influence the impact of IPTp on haemoglobin that are not captured in our framework, and we have now made this limitation more explicit in the Discussion.

In addition, one point remains only indirectly addressed: my suggestion to explore whether areas with higher malaria prevalence correspond to higher inferred immunity levels in primigravidae. While the additional age-gravidity analysis is valuable, it does not directly test this spatial consistency, which could still provide a useful validation of the model's inferred immunity component, even if only descriptively.

We apologise for not addressing this point more clearly, we agree this is an important consideration. The relevant empirical pattern is shown in Figure 1c (G1 panel), which demonstrates that the reduction in Hb associated with infection among primigravidae does not display a significant relationship with transmission intensity across our study sites.

We have now pointed to this more clearly in the text and expanded upon in our discussion with the following:

“Meanwhile, our findings that the impact of malaria in primigravid women did not vary significantly across study sites and that pregnancy-specific immunity acquired from exposure during previous pregnancies, rather than that acquired by exposure to malaria prior to first pregnancy, is the prime modifier of malaria impact on Hb during pregnancy. However, this does not preclude a much more prominent role of non-pregnancy acquired immunity in lower transmission settings than assessed here (i.e. <10% PCR prevalence). Given these are settings where exposure is increasingly rare, such effects may not substantially alter our overall burden estimates but may have important consequences for individuals affected and highlight the importance of maintaining protection through accurate diagnosis of infection

early in pregnancy and the exercise of caution when determining transmission thresholds below which IPTp should not be considered.”

Reviewer #2 (Remarks to the Author):

The authors have responded to my queries in a satisfying manner, and I think the manuscript is acceptable for publication.

We thank the reviewer for this encouraging assessment and for their constructive input throughout the review process, which has substantially improved the manuscript.